

# Modelling potential production and environmental effects of macroalgae farms in UK and Dutch coastal waters

Johan van der Molen[1,2], Piet Ruardij[2], Karen Mooney[4], Philip Kerrison[5], Nessa E. O'Connor[4], Emma Gorman[4], Klaas Timmermans[3], Serena Wright[1], Maeve Kelly[5], Adam D. Hughes[5], Elisa Capuzzo[1]

[1]The Centre for Environment, Fisheries and Aquaculture Science (Cefas), Lowestoft, NR33 0HT, UK
[2]NIOZ Royal Netherlands Institute for Sea Research, Dept. of Coastal Systems and Utrecht University, Den Burg, 1797 SZ, The Netherlands
[3]NIOZ Royal Netherlands Institute for Sea Research, Dept. of Estuarine and Delta Systems and Utrecht University, Yerseke, 4401 NT, The Netherlands
[4]Queen's University, Belfast, BT7 1NN, UK
[5]The Scottish Association for Marine Science (SAMS), Oban, PA37 1QA, UK

*Correspondence to*: Johan van der Molen (johan.vandermolen@cefas.co.uk, johan.van.der.molen@nioz.nl)

**Abstract.** There is increasing interest in macroalgae farming in European waters for a range of applications, including food, chemical extraction and as biofuels. This study uses a 3D numerical model of hydrodynamics and biogeochemistry to investigate potential production and environmental effects of macroalgae farming in UK and Dutch coastal waters. The model included four experimental farms in different coastal settings in Strangford Lough (Northern Ireland), in Sound of Kerrera and Lynn of Lorne (northwest Scotland), and in the Rhine Plume (The Netherlands), as well as a hypothetical large-scale farm off the UK north Norfolk coast. The model could not detect significant changes in biogeochemistry and plankton dynamics at any of the farm sites averaged over the farming season. The results showed a range of macroalgae growth behaviours in response to simulated environmental conditions. These were then compared with *in-situ* observations where available, showing good correspondence for some farms and less good correspondence for others. At the most basic level, macroalgae production depended on prevailing nutrient concentrations and light conditions, with higher levels of both resulting in higher macroalgae production. It is shown that under non-elevated and interannually varying winter nutrient conditions, farming success was modulated by the timings of the onset of increasing nutrient concentrations in autumn and nutrient drawdown in spring. Macroalgae carbohydrate content also depended on nutrient concentrations, with higher nutrient concentrations leading to lower carbohydrate content at harvest. This will reduce the energy density of the crop and so affect its suitability for conversion into biofuel. For the hypothetical large-scale macroalgae farm off the UK north Norfolk coast the model suggested high, stable farm yields of macroalgae from year to year with substantial carbohydrate content and limited environmental effects.



# 1    Introduction

## 1.1    Background, aims and approach

Worldwide macroalgae (seaweed) production is in excess of 28 million ton p.a. and has doubled between 2000 and 2014 (FAO, 2014). The majority of this production (> 95%) is from the SE Asian region where macroalgal cultivation is well established (FAO 2014, West et al., 2016). The harvested macroalgae biomass is mainly used directly for human consumption, although other uses include the extraction of phyllocolloids (gelling agents), animal feed, fertilizer, water remediation and as probiotics in aquaculture (see van der Burg et al. 2016; West et al. 2016).

There has been increasing interest in the potential of macroalgae cultivation across the Northern Hemisphere and Europe (van der Burg et al., 2016), partially driven by research on biofuel technologies (Kerrison et al., 2015). The characteristics of Phaeophyta macroalgae, in particular high productivity, fast growth rate and high polysaccharide content, make them a suitable biomass for biofuels production (Hughes et al., 2012; Kerrison et al., 2015; Schiener et al., 2016; Fernand et al., 2017). A further advantage is that such third generation biofuels do not need additional freshwater and do not compete for agricultural land like many existing biofuel sources.

Marine macroalgae fix $CO_2$, acting as a sink for anthropogenic $CO_2$ ("Blue Carbon", Nellemann et al., 2009; Duarte et al., 2017) and absorb dissolved nutrients from the water column, helping to remediate nutrient release from anthropogenic sources such as agricultural runoff, waste water treatment and aquaculture ('bioremediation', e.g. Fei, 2004; He et al., 2008; Chopin et al., 2001; Lüning and Pang, 2003; Sanderson et al., 2012; Smale et al., 2013). Therefore, large-scale cultivation and harvesting of macroalgae could play a role in removing carbon from the marine environment, as well as reduction of coastal nutrient enrichment.

Kelp species, such as *Saccharina latissima* (a brown algae), have been identified as candidate macroalgae for bioenergy production (Kerrison et al., 2015). Its cultivation has been trialled across Europe, including Scotland, Strangford Lough in Northern Ireland, southern North Sea, and northwest of Spain (Kerrison et al. 2015; Buck and Buchholz 2004; Sanderson et al. 2012; Peteiro et al. 2012; van der Burg et al. 2016).

Kelp naturally occurs in sublittoral coastal waters in temperate and polar regions. These macroalgae aggregations have been shown to modify the surrounding environment by reducing water velocity and attenuate waves (Gaylord et al., 2007; Jackson, 1997), and by modifying sedimentation rates of suspended particles (Eckman et al., 1989). They are also associated with high biodiversity (Burrows, 2012), providing numerous ecosystem services including habitat, shelter and food for many species including fish (Hartney, 1996), benthic organisms (lobster, crabs; Bologna and Steneck, 1993; Daly and Konar, 2008), herbivorous organisms (Kang et al., 2008), and birds (Fredriksen, 2003), and are associated with high biodiversity (Burrows, 2012), see also Walls et al. (2017).



While a large scale kelp farm might replicate some of the ecosystem services of a natural kelp forest, assumptions as to the extent of the similarity should be considered with caution (Wood et al., 2017). Since kelp farms are monocultures suspended within the water column, and are likely to undergo a yearly cycle of growth and harvesting, they are not synonymous with mature kelp beds which contain a mixture species, of different ages attached to the benthos (Wood et al. 2017).

5   Studies on the potential environmental effects of macroalgae farms are limited. This lack of information, in combination with limited knowledge on expected farm yields, results in uncertainty for potential investors, developers and macroalgae farmers, as well as legislators, who provide the relevant farming licence (Wood et al., 2017).

The aim of this modelling study was to investigate environmental effects and potential yield of macroalgae farms, at different locations in UK and Dutch coastal waters, using the ERSEM-BFM (European Regional Seas Ecosystem Model - Biogeochemical Flux Model). In particular, four farms were simulated: three experimental farms (Sound of Kerrera, Scotland; Strangford Lough, Northern Ireland; the Rhine region of fresh-water influence, ROFI, The Netherlands) and a hypothetical farm (Norfolk, UK). Observations from the experimental farms in Scotland and Northern Ireland were used to ground truth the model.

This modelling exercise is a proof of concept, and did not aim for a detailed representation of the farm localities, nor did it involve extensive tuning to reproduce detail of farm performance. We used an existing 3D model setup of the northwest European continental shelf (Section 2.2), which allowed all farms to be included in one model, albeit with a very coarse representation of coastal geometries. Farm implementation included a level of sub-grid parameterisation. The model was run with forcings for years pre-dating farm deployments, so comparisons with observations collected during the actual deployments can only be qualitative. Despite these limitations, we obtained reasonable confidence in the model, as well as valuable results in terms of farm functioning and performance, macroalgae quality, and farming-induced changes in environmental conditions. These predictions are necessary to progress the future development of this fledgling industry.

## 1.2    Study area

### 1.2.1    Southern North Sea

Two of the sites were located in the southern North Sea. The North Sea (Figure 1) is a shallow shelf sea of depths up to 600 m, but typically averaging only several tens of metres in the south. The glacial ice-pushed ridge that forms Dogger Bank separates the southern area from the central and northern North Sea where average depths are ca. 100 m. The Norwegian Trench and Skagerrak separate these areas from the Scandinavian coast with depths of several hundreds of metres.

An overview of the hydrography of the North Sea was compiled by Otto et al. (1990). The tides are predominantly semi-diurnal, with ranges of up to several metres along the coasts, and amphidromic points of the main $M_2$ tidal constituent in the Southern Bight, the German Bight and near the southern tip of Norway (Proudman and Doodson, 1924; Prandle, 1980).



Model-based estimates of a range of tidal constituents were derived by Holt et al. (2001), and also Davies et al. (1997). Tidal currents can reach speeds of over 1 ms$^{-1}$ in coastal areas (Dietrich, 1950; Holt et al., 2001).

The central and northern North Sea stratify in summer, while southern area remains well-mixed (Pingree et al., 1978; Van Leeuwen et al., 2015). The residual circulation is generally clock-wise, with inflows along the Atlantic boundary in the North and through the Strait of Dover (North Sea Task Force, 1993), but with seasonal variations (Holt et al., 2001). Contributions by wind and tides are of comparable magnitude (Prandle, 1978). During stratified conditions in summer, subsurface jet-like currents occur near the thermocline around the Dogger Bank (Brown et al., 1999; Hill et al., 2008). Tracing of accidental radioactive releases in the 1970s indicated that it takes several years for waters to traverse the region (Kautsky, 1973).

The North Sea supports a high level of primary productivity, which is augmented by varying levels of anthropogenic riverine nutrient loads, which have been gradually reducing since 1985. This phytoplankton production is also modulated by turbidity and light availability, influenced by among others the concentration of suspended particulate materials (SPM) in the water (e.g., Lenhart et al., 2010). The anthropogenic nutrient load is typically contained within coastal plumes that propagate in a clockwise direction along the coasts (Menesguen, 2006; Painting et al., 2013; Los et al., 2014; Desmit et al., 2015). Potential oxygen depletion as a result of eutrophication has received attention recently, focusing in particular on the German Bight (Topcu and Brockmann, 2015; Grosse et al., 2016).

The north Norfolk coast of the UK and the southern coast of the Netherlands, where two of the farms under investigation were located, are characterized by shallow water depths (<25 m), high winter nutrient concentrations (Hydes et al., 1999; Proctor et al., 2003; Foden et al., 2011; Laane, 2005; Troost et al., 2014), and high turbidity (Dyer and Moffatt, 1998; Bristow et al., 2013; Pietrzak et al., 2011; Van der Hout et al., 2015). In contrast to the north Norfolk coast, the Dutch coastal area has lower salinity, potential for episodic salinity stratification (De Ruyter et al., 1997), and a higher N/P ratio due to larger reductions in anthropogenic riverine phosphate loading since the late 1980's (Lenhart et al., 2010).

### 1.2.2    Strangford Lough (Northern Ireland)

The Northern Irish farm site run by Queen's University, Belfast, is located at 54.4N, 5.58W within the semi-enclosed Strangford Lough. The Lough covers an area of approximately 134 km$^2$, with water depths from 0-70 m, is 8 km long and the Narrows (0.5 km wide at narrowest) connect the Irish Sea to the main inlet of the lough (Smyth et al. 2016; Kregting and Elsäßer, 2014). The lough is fully saline ranging from 32 to 34 with negligible freshwater input from three small point sources (Boyd, 1973; Smith, 2010) and is predominantly well mixed (Taylor and Service,1997). The experimental site is located off the southern shore in the vicinity of the Narrows, but is relatively sheltered with an average current speed of 0.3 ms$^{-1}$. There can be moderate wave action when the wind is coming from northerly and easterly directions. The depth profile



is variable, ranging from 2 m to 13 m at Mean Low Water Spring (MLWS). The current predominantly runs in a west – east direction (Mooney-McAuley et al. 2016).

### 1.2.3     Sound of Kerrera and Lynn of Lorne (Northwest Scotland)

The first Scottish farm site was located at 56.38N, 5.54W within the Sound of Kerrera which separates the island Kerrera from the mainland by ca. 500m, near Oban. The Sound reaches 60m depth and experiences a semi-diurnal tidal current of 0.77 m·s$^{-1}$ during spring tides. The island shelters the Sound from all but the predominant south-westerly winds from the Atlantic. At the farm site, the depth ranges from 5-25m.

The second Scottish farm is located at 56.49N, 5.47W in the Lynn of Lorne, which separates the island of Lismore from the mainland. The site range from 15-25 m depth and has a mean current speed of 0.1 m·s$^{-1}$, 5 m below the surface. The Lynn of Lorne is 3km wide at the location of the farm and so is very exposed to the predominant south-westerly winds from the Atlantic.

### 1.3     *Saccharina latissima*

*Saccharina latissima*, or sugar kelp, is a subtidal phaeophyte macroalga native to Europe, common to UK rocky shores. It is a brown algae, with leather/rubbery texture, which in the adult form is constituted by a holdfast, a stipe, and a large undivided blade (or frond, or lamina), with undulated margins (Kain 1979; White and Marshal 2007).

The growth of *S. latissima* is affected by environmental factors such as light availability, wave action and water currents, nutrient concentration, type of substratum, temperature, salinity and grazing pressure (Birkett et al. 1998; Lobban and Harrison 1997). A recent study by Kerrison et al. (2015) summarises the optimal range of environmental variables for *S. latissima* growth (see Table 1 in Kerrison et al. 2015).

In coastal waters around the UK, kelp species show high growth rates from late autumn to early summer. This is then followed by a slower growth phase between July and December (Parke 1948; Kain 1963). Maximum length developments are also associated with maximum fresh weights (Parke 1948; Black 1950).

Kelp plants show effective uptake of nutrients (nitrogen and phosphate) from seawater (Birkett et al. 1998; Kregting et al., 2014; Kregting et al., 2016). When nutrients are abundant and exceed metabolic requirements, these plants have the ability to store nutrient in the plant tissues (Birkett et al. 1998). For example, *S. latissima* has been shown to store nitrogen reserves at levels of more than 1000 times the external ambient concentration (Chapman et al. 1978).



*S. latissima* stores energy in the form of carbohydrates (e.g. mannitol and laminarin), the concentrations of which vary widely during the year, and peak in the second part of the year (Black 1950; Kain 1979; Bartsch et al. 2008; Schiener et al., 2015). For example, Gevaert et al. (2001) observed that for *S. latissima* in the English Channel, the maximum carbon content is reached in September with the lowest concentrations occurring in March. Similar trends have also been reported for

Norway (Sjøtun 1993) and Scotland (Connolly and Drew 1985). The minimum carbon concentration in March occurs when the growth rate of the algae is high and the plant growth is carried out at the expense of carbohydrate reserves. By contrast during summer, carbon assimilation exceeds carbon-utilisation allowing the formation of carbon reserves (Gevaert et al. 2001).

The presence of these carbohydrate reserves and a fast growth rate make *S. latissima* an interesting potential biomass for production of renewable energy (Kraan, 2013; Fernand et al., 2017). For these reasons, *S. latissima* has become a focus for experimental farming in Europe. For a comprehensive summary of modelling efforts on *S. latissima* we refer the reader to Broch & Slagstad (2011) and references therein.

**2    Methods**

### 2.1    Macroalgae farms

### 2.1.1    Strangford Lough research farm

The Strangford Lough research farm is located near the southwestern shore of the Lough (Figure 1, farm A), and run by Queen's University, Belfast. The farm cultivated a mixture of *S. latissima*, *Laminaria digitata* and *Alaria esculenta*. Here we

use observations from the 2012-2013 deployment, when 2 x 100m longlines of *S. latissima* were cultivated and 19 with the other species. The growing lines were suspended horizontally at 1 m below the surface, and were pre-seeded at deployment. Monthly sampling was carried out after two months at sea. At each sampling time, five samples were taken from each rope removing all plants on a 30 cm section. The total wet biomass of all plants in these 30 cm intervals was determined, and used to calculate the mean biomass per line (kg wet weight/m). Total number of plants, total wet and dry biomass were measured.

Total length, blade length, blade width and stipe length were also measured for the 12 largest plants in the sample. For the purpose of this modelling study, we have assumed that all 21 lines of the farm were cultivated with *S. latissima*.

The mean biomass per line (kg wet weight/m) was used to estimate the total farm carbon biomass for comparison with the model results (see Section 3.3). Overall, 354 samples were analysed for wet and dry weight, giving a combined total wet

weight of 3549.9 kg, and a dry weight of 380.38 kg, resulting in a wet/dry weight ratio of 9.333. For conversion from dry weight to carbon weight we assume that the dry plant material consists predominantly of $CH_2O$ groups, resulting in a dry weight to carbon ratio of 32/12=2.67.



### 2.1.2    Sound of Kerrera and Lynn of Lorne research farms

The Sound of Kerrera and Lynn of Lorne farms (Figure 1, farms B and C) are located in the Firth of Lorne, and operated by the Scottish Association for Marine Science (SAMS), see also Section 1.2.3. The farm at Sound of Kerrera (Figure 1, B) consists of 180 m of double headed longline buoyed by mussel floats, with growing lines suspended at 1.5 m depth. The Lynn of Lorne farm (Figure 1, C) consists of a 100x100 m grid submerged 3 m below the surface. This can contain up to 24 lines of 100 m length, spaced 4 m apart, with growing lines suspended horizontally at a depth of 1.5 m below the surface. For the Sound of Kerrera farm, observations of nutrient concentrations, light and temperature are available from a 17 month period in 2013-14. Nutrient concentrations were collected in triplicates at 1.5 m depth, whereas light and temperature were collected at half hourly intervals at 1.5 m depth, using HOBO Pendant data loggers (Onset Computer Corp, MA, USA). Here, we use the means of the triplicates for nutrients, and monthly means for light and temperature. The nutrient data showed a typical seasonal cycle with high winter concentrations and low concentrations following the spring bloom, but with surprisingly high summer concentrations in 2013, which are unexplained. Early summer concentrations in 2014 were substantially lower. Typical macroalgae yields for Sound of Kerrera were ca. 10 kg wet weight $m^{-1}$ in 2013, and 4 kg wet weight $m^{-1}$ in 2014.

### 2.1.3    Rhine plume experimental farm

Another experimental farm, run by North Sea farm foundation (Stichting Noordzee boerderij), was deployed for the first time in the autumn of 2016 within the nutrient-rich Rhine Region of Fresh-water Influence (ROFI) off the port of Scheveningen, The Netherlands (Figure 1, D). The farm consists of a single line of 100 m, undulating between 0 and 4 m below the surface. Data from this farm will only become available in the summer of 2017. The farm was included in the model to obtain predictions of potential performance.

### 2.1.4    Norfolk hypothetical commercial farm

The hypothetical commercial farm off north Norfolk was selected based on the method of Capuzzo et al. (2014), with minor modifications. The method consisted of over-laying maps of suitability scores (optimal, sub-optimal, unsuitable) of key limiting environmental variables (temperature, light, tidal velocity, wave height and nutrient concentrations; Table 1) and spatial use data (shipping, structures, MPA's, wind farms, etc.) in a GIS system. The modifications applied here consist of slight variations to the threshold levels of certain environmental variables, and the adoption of a farming area based on the suitability data rather than rectangles of pre-defined size.

The area selected by this method was nearly rectangular (53.0545N 0.7745E to 53.11N 0.9775E), and located off the north Norfolk coast near Wells-next-the-Sea, in approximately 20 m water depth, between a coastal Marine Protected Area, wind farms and a Marine Conservation Zone further offshore (Figure 2). On the model grid, this area was approximated by a



hypothetical farm covering three adjacent cells of 0.08 degree longitude and 0.05 degree latitude (approx. 5x5 km, Figure 1, E-G).

It was assumed that within each grid cell of 25 km$^2$, roughly half of the surface area would be effectively farmed, and the rest would be required for a mesh of navigation corridors for service vessels and occasional navigation lanes for other traffic. As details of such lay-outs are beyond the resolution of the model, it was assumed for simplicity that a solid block of 3.5x3.5 km was farmed within each 5x5 km grid cell with lines 50 m apart to avoid entanglement.

## 2.2    GETM-ERSEM-BFM model

### 2.2.1    GETM: North-west European Shelf set-up

The 3D hydrodynamic model GETM (General Estuarine Transport Model, www.getm.eu; Burchard & Bolding, 2002) solves the shallow-water, heat balance and density equations. It uses GOTM (General Ocean Turbulence Model, Burchard et al., 1999; www.gotm.net) to solve the vertical dimension. GETM was run using the north-west European shelf setup that has been used by Van der Molen et al. (2016) to study the potential large-scale effects of tidal energy generation in the Pentland Firth, and by Van der Molen et al. (2017) to develop a suspended particulate matter model. The set-up includes a spherical grid covering the area 46.4°N-63°N, 17.25°W-13°E with a resolution of 0.08° longitude and 0.05° latitude (approximately 5.5 km), and 25 non-equidistant layers in the vertical. The model bathymetry was based on the NOOS bathymetry (www.noos.cc/index.php?id=173). The model was forced with tidal constituents derived from TOPEX-POSEIDON satellite altimetry (LeProvost et al., 1998), atmospheric forcing from ECMWF ERA-Interim (Dee et al., 2011; Berrisford et al., 2011; www.ecmwf.int/en/research/climate-reanalysis/era-interim), interpolated river runoff from a range of observational data sets (the National River Flow Archive (www.ceh.ac.uk/data/nrfa/index.html) for UK rivers, the Agence de l'eau Loire-Bretagne, Agence de l'eau Seine-Normandie and IFREMER for French rivers, the DONAR database for Netherlands rivers, ARGE Elbe, the Niedersächsisches Landesamt für Ökologie and the Bundesanstalt für Gewässerkunde for German rivers, and the Institute for Marine Research, Bergen, for Norwegian rivers; see also Lenhart et al., 2010), and depth-resolved temperature- and salinity boundary conditions from ECMWF-ORAS4 (Balmaseda et al., 2013; Mogensen et al., 2012; http://www.ecmwf.int/products/forecasts/d/charts/oras4/reanalysis/). Boundary conditions for nutrients are taken from the World Ocean Atlas monthly climatology (Garcia et al., 2010).

### 2.2.2    Macroalgae farms in ERSEM

The ERSEM-BFM (European Regional Seas Ecosystem Model - Biogeochemical Flux Model) version used here (01-06-2016) is a development of the model ERSEM III (see Baretta et al., 1995; Ruardij and Van Raaphorst, 1995; Ruardij et al., 1997; Vichi et al., 2003; Vichi et al., 2004; Ruardij et al., 2005; Vichi et al., 2007; Van der Molen et al., 2013; van der Molen et al., 2014; van der Molen et al., 2016; www.nioz.nl/northsea_model), and describes the dynamics of the biogeochemical fluxes within the pelagic and benthic environment. The ERSEM-BFM model simulates the cycles of carbon,



nitrogen, phosphorus, silicate and oxygen and allows for variable internal nutrient ratios inside organisms, based on external availability and physiological status. The model applies a functional group approach and contains five pelagic phytoplankton groups, four main zooplankton groups and five benthic faunal groups, the latter comprising four macrofauna and one meiofauna groups. Pelagic and benthic aerobic and anaerobic bacteria are also included. The pelagic module includes

transparent exopolymer particles (TEP) excretion by diatoms under nutrient stress, the associated formation of macro-aggregates consisting of TEP and diatoms, leading to enhanced sinking rates and food supply to the benthic system especially in the deeper offshore areas (Engel, 2000), a *Phaeocystis* functional group for improved simulation of primary production in coastal areas (Peperzak et al., 1998), a pelagic filter-feeder larvae stage, and benthic diatoms, including resuspension, transport and pelagic growth. The suspended particulate matter (SPM) module, included for improved

simulation of the under-water light climate, contains contributions by waves and currents, and full 3D transport (van der Molen et al., 2017). Finally, the model includes resuspension of particulate organic matter as a proportion of the SPM resuspension, and also 3D transport.

A macroalgae functional type representing *Saccharina latissima* was introduced in ERSEM-BFM, closely following the

implementation of Broch and Slagstad (2012), but with addition of phosphate dynamics in analogy to the nitrate diynamics, and assuming an optimum N/P ratio for structural mass of 25, slightly below the median reported by Atkinson & Smith (1983). The nutrient uptake method for the macroalgae was changed to the dynamic one presented by Droop (1973, 1974) in order to be consistent with the nutrient uptake of phytoplankton in ERSEM (Baretta-Bekker et al., 1997). See Figure 3 for a schematic diagram of the implementation. The method includes growth, mortality ('erosion'), nutrient and carbon

biogeochemistry, and effects of light, temperature, and nutrient concentrations. Plant structural biomass, nutrient buffers and carbohydrate biomass were represented separately. For further detail see Broch and Slagstad (2012). For inclusion in ERSEM-BFM, the macroalgae were represented in terms of biomass density rather than frond dimensions. Only farmed macroalgae were included in the model. The implementation of farms assumed the use of lines as an anchoring material. Farms were prescribed, per model grid cell, in terms of line length, number of lines, depth below the surface, deployment

and harvest time, and initial biomass and plant density (see Table 2 for detail). The simulated farms coinciding with the experimental farms in Strangford Lough, Sound of Kerrera, Lynn of Lorne and the Rhine Plume were given dimensions coinciding with typical deployments. The background of the dimensions of the north Norfolk farm was given in Section 2.1.4. To facilitate the comparisons, all simulations used the same deployment and harvest dates.

### 2.2.3    Model scenarios

GETM-ERSEM-BFM was run without macroalgae farms from 1990 to 2011, using initial conditions from an earlier model version. The first 10 years of this simulation were considered as spinup time to enable the biogeochemistry of the model to adjust. The years 2001-2011 constituted the reference conditions (absence of farms). Farming scenarios were run for five consecutive seasons, starting on the first of October in 2006-2010, and running until the end of July of the following year.



The scenario runs were hot-started for each year from the corresponding conditions of the reference run on 1 October. To detect potential environmental effects, differences with the reference run were calculated of farm-season averaged model scenario output for all routinely stored variables, filtered for model variability using the method of Van der Molen et al. (2016), and plotted as maps. Time series consisting of daily values were extracted for pelagic nutrients, light conditions and

macroalgae conditions at each model grid cell containing a macroalgae farm to assess farm performance and functioning.

### 2.3    SmartBuoy and satellite observations

SmartBuoys, instrumented moorings (Mills et al., 2005) have been deployed in UK and Dutch waters as components of monitoring programmes, and were configured to determine turbidity, chlorophyll fluorescence, salinity, temperature and dissolved oxygen and data processed according to Greenwood et al. (2010). Concentrations of suspended particulate matter

and chlorophyll were derived from measurements of turbidity and chlorophyll fluorescence respectively (Greenwood et al., 2010). Discrete samples were collected using an automated Aquamonitor and subsequently analysed for TOxN (total oxidisable nitrogen) and silicate according to Gowen et al. (2008). In addition on most buoys, TOxN was determined using an automated *in-situ* NAS-2E or NAS-3X nutrient analyser. Daily mean values were calculated from all data which passed the quality assurance process.

Daily spatial distributions of chlorophyll concentrations were derived from the MODIS satellite (modis.gsfc.nasa.gov), obtained from the Ifremer ftp server (ftp.ifremer.fr.:/ifremer/cersat/products/gridded/ocean-color/atlantic, and which were processed as described by Gohin et al. (2005) and Gohin (2011)). These data were further processed, in conjunction with modelled surface chlorophyll concentrations, to yield spatially resolved summer and winter statistics of model performance.

## 3    Results

### 3.1    Model confirmation

Modelled $M_2$ tidal elevations and currents were compared with observations by Van der Molen et al. (2016), showing reasonable agreement, with elevation amplitudes typically within 20 cm, currents typically within 15 cm s$^{-1}$, and phases for both typically within 30°. Compared with *in-situ* SmartBuoy observations (Greenwood et al., 2010), modelled SPM

concentrations showed a reasonable representation of the seasonal cycle, but over-estimating peak values. They were mostly within a factor of three, and with positive correlations, when compared with satellite observations on a seasonal scale (Van der Molen et al., 2017). These results are not reproduced here.

Surface chlorophyll concentrations were compared with satellite observations for 2007-2008 (see also Van der Molen et al.,

2016). Winter concentrations (Figure 4a,b) were low in both the model and the satellite data. For a better comparison, the model output was subsampled for each grid cell (Figure 4c) using the available clear-skies satellite observations (Figure 4d). Subsequently, the relative offset (Figure 4e) and correlation coefficient (Figure 4f) were calculated. The resulting plots show



that the model over-predicted in the Atlantic Ocean, in the North Sea along the northern UK coast, and in the Norwegian Trench. Correlations showed a patchy pattern, with typically better correlations in the northern North Sea and along the continental coast.

In summer (Figure 5) the model had a small bias in offshore waters in the North Sea and English Channel, but tended to over-estimate coastal and oceanic chlorophyll concentrations. It achieved good correlations in large parts of the North Sea and on the south-western shelf.

In the vicinity of the North Norfolk farm, the model bias for surface chlorophyll was slightly negative in winter, and the
correlation coefficient was low (Figure 4e,f). In summer, chlorophyll concentrations were slightly over-estimated, and correlations were moderate (Figure 5e,f). Near the Rhine Plume farm, bias was slightly positive and correlations high in winter (Figure 4e,f), and bias was slightly positive and correlations moderate in summer (Figure 5e,f).

The model results were compared with time series of *in-situ* observations from SmartBuoy for chlorophyll, nitrate, silicate,
salinity, temperature and suspended sediment. For Warp Anchorage (see Figure 1 for location) peak spring-bloom chlorophyll concentrations were within 10 mg Chl m$^{-3}$ for most years (Figure 6a). The blooms tended to have longer duration than observed. Winter nitrate and silicate concentrations exceeded observed values for most years (Figure 6b,c), and were related to lower salinity values than observed (Figure 6d). The modelled annual range in temperatures was several degrees more than observed (Figure 6e), and suspended sediment concentrations were much more variable, and had high event-
driven peak values (Figure 6f).

In Liverpool Bay (see Figure 1 for location), spring and summer chlorophyll concentrations generally exceeded observed values from the SmartBuoy by a factor of two (Figure 7a). Nitrate concentrations were reproduced well in the last five years of the simulation, but were over-estimated in the first four winters (Figure 7b). Winter silicate concentrations were also
higher in the first few years, but exceeded observed winter values for all the years in the time series (Figure 7c). Modelled salinities were slightly higher than observed (Figure 7d), and there was no apparent relationship with winter nutrient concentrations as for Warp Anchorage. Summer temperatures were reproduced mostly within a degree, while winter temperatures were underestimated by up to 2 °C (Figure 7e). The seasonal cycle of SPM concentrations was reproduced, but with substantially higher variability (Figure 7f).

At the more offshore location of West Gabbard (see Figure 1), peak chlorophyll concentrations were underestimated for most, but not all of the years (Figure 8a). Nitrate concentrations were under-estimated by a factor of 2-3 (Figure 8b), whereas silicate concentrations were reproduced fairly closely (Figure 8c). Summer salinities were over-estimated by 0.8-1.2 (Figure 8d). Maximum summer temperatures were exceeded by up to 2 °C in most years, and minimum winter temperatures were,



with a few exceptions, reproduced closely (Figure 8e). Winter suspended sediment concentrations were 4-5 times higher than observed, with much higher variability (Figure 8f). This general pattern was also observed at other offshore SmartBuoys (not shown here for brevity).

## 3.2    Environmental effects

None of the maps of differences in biogeochemistry and plankton dynamics with the reference run, averaged over the farming season, showed detectable changes in the region of any of the farm sites, i.e. any differences were smaller or of similar magnitude as differences between the two reference runs. For the experimental farms, this was to be expected because of their relatively small size. The north Norfolk farm was located in a dynamic area with high tidal currents and substantial residual circulation, which may account for this result. Hence, in the following, we will focus on the performance
of the macroalgae farms.

## 3.3    Strangford Lough

Modelled winter nutrient concentrations at the Strangford Lough farm site (Figure 9a,b), 1.5-5 mmol N m$^{-3}$ and 0.16-0.3 mmol P m$^{-3}$, showed substantial variation between years, and were lower than expected for a coastal location. Reported values for the Narrows and coastal offshore area for 2009 show values within, but also in exceedence of these ranges
(Kregting et al., 2016). One reason for this could be that nutrient inputs from the largest river entering Strangford Lough, the Quoile, were not available for inclusion in the model. Summer concentrations were close to zero, with a suggestion that nitrogen was the limiting nutrient. Extinction coefficients (Figure 9c) ranged from peak winter values of up to 3 m$^{-1}$ to summer values of 0.2-0.3 m$^{-1}$ with fairly similar seasonal patterns per year. Surface irradiance (Figure 9d) showed a typical and stable seasonal cycle ranging from around 10 μmol m$^{-2}$ s$^{-1}$ in winter to maxima of 800 μmol m$^{-2}$ s$^{-1}$ in summer. Water
temperatures (Figure 9e) ranged from about 16 °C in summer to 5-7 °C in winter, with the winters of 2008-2010 slightly colder than those of 2006-2007. Macroalgae biomass (Figure 9f) peaked at approximately 0.5 kg C m$^{-1}$ line in all simulated years, about an order of magnitude lower than observed in the 2012-2013 farm deployment (plotted here in 2006). This under-estimation was caused by the low nutrient concentrations, as is also evident in the structure/mass ratio (Figure 9g) and C/N and C/P ratio (Figure 9i,j), which show that the modelled macroalgae was high in carbohydratate content from the
beginning and then rose throughout the cultivation. Mortality (Figure 9h) remained low throughout the farming cycles. Nutrient uptake rates per surface area of the farm (Figure 9k,l) were close to zero in summer, and peaked between 0.05-0.15 mmol N m$^{-2}$ day$^{-1}$ and 0.015-0.025 mmol P m$^{-2}$ day$^{-1}$.

## 3.4    Sound of Kerrera and Lynn of Lorne

For the Sound of Kerrera farm (Figure 10), winter nutrients were higher, but also with substantial differences between years
(7-15 mmol N m$^{-3}$ and 0.6-1.4 mmol P m$^{-3}$). This is closer to expected values than at Strangford Lough, which is also





illustrated by the observations from 2013-14, here plotted on 2010-11. The model did not reproduce the high summer concentrations evident in the first year of the observations. Extinction coefficients had similar values as in Strangford Lough, but with a lower base level in winter. Irradiance was also similar, and corresponded well with the monthly mean observed values from 2013-14. Water temperatures reached up to 18 °C in summer, and 3-6 °C in winter, ranges confirmed by the

monthly mean observed values from 2013-14. Macroalgae biomass at harvest showed substantial interannual variability, between 0.11 and 0.48 kg C m$^{-1}$ line. This range corresponds with the observed farm yield of 0.4 kg C m$^{-1}$ line in 2013 and 0.16 kg C m$^{-1}$ line in 2014, and also with the observed yield of 0.6 kg C m$^{-1}$ line for the Strangford Lough farm in 2012/13. While the observed difference in yield seems to correspond with the observed difference in summer nutrient concentrations, the farm operators also reported heavy biofouling in 2014, which smothered the crop, possibly due to warmer spring

temperatures in this particular year. The modelled differences in yield appear to relate to the nitrogen uptake rates. The final modelled carbohydrate content was high. Rates of mortality increased with biomass.

The Lynn of Lorne farm (Figure 11) showed a very similar pattern, but achieved slightly higher yields due to higher modelled nutrient concentrations. Interestingly, the carbohydrate content at harvest was lower for the high-yield years. In the

year with highest yield, mortality shot up ten-fold shortly before harvest, suggesting that timing of harvesting may be critical. This latter result corresponds with the experience of the farm operators.

### 3.5    Rhine plume experimental farm

For the Rhine plume farm (Figure 12), the model over-predicted winter nitrate concentrations as compared with observations from the Noordwijk-10 station further to the north by up to a factor of three (up to 180 mmol N m$^{-3}$), and also over-predicted

summer concentrations. Phosphate concentrations were reproduced fairly closely, and both model and observations suggest that for a period after the spring bloom phosphate was the limiting nutrient. The model also reproduced the available observations of the extinction coefficient, which bottomed out at approximately 0.5 m$^{-1}$; higher than at the Scottish and Irish farm sites. Summer temperatures ranged up to 20 °C, which is near the thermal limit for this kelp species. Farm yields were relatively stable at around 0.7 kg C m$^{-1}$ line, but carbohydrate content remained low as nutrients remained available

throughout the summer. The observations suggest that this may not be as extreme in reality. Mortality increased during the later stages of growth, suggesting that if, in reality, the carbohydrate content does increase due to lower nutrient concentrations than in the model, there may be a fine balance in picking the right time to harvest.

### 3.6    Norfolk hypothetical commercial farm

The results for the Norfolk hypothetical farm (Figure 13) showed winter nitrate concentrations of 40-50 mmol N m$^{-3}$ and 1.5-

2.5 mmol P m$^{-3}$, respectively. This corresponds with observed values of 45-48 mmol N m$^{-3}$ and DIN/DIP ratios of 20-30 for the Eastern English Coast and East Anglia regions in 2001-2005 (Foden et al., 2011). Extinction coefficients were over-estimated by the model by a factor of 3-4. This was compensated for by setting the farm lines to 0.3 m below the surface





instead of 1 m. Summer temperatures ranged up to 20 °C, while winter temperatures could be as low as 2.5 °C. Farm yields were stable and high at approximately 1 kg C m$^{-1}$ line, and the final crop contained substantial concentrations of carbohydrates. Mortality did not increase as much as at some of the other sites.

### 3.7    Predicted farm yields

In addition to the per unit performance of the farms presented in the previous section, it is, from the point of view of biomass production, useful to list the total predicted yield of the farms at their current size. Total modelled farm yields are summarised in Table 3. In terms of wet biomass, yields were in the range of 2-3 t yr$^{-1}$ for the Strangford Lough farm, 7-30 t yr$^{-1}$ for the Sound of Kerrera farm, 20-60 t yr$^{-1}$ for the Lynne of Lorne farm, around 1.5 t yr$^{-1}$ for the Rhine plume farm, and 18 and 20 kt yr$^{-1}$ for the combined Norfolk farm.

### 3.8    Variations in farm yield

The model results suggested that macroalgae growth was dictated by combined availability of nutrients and a sufficient level of light. To illustrate this, nutrient uptake was plotted as a function of irradiance and nutrient concentration. The resulting graphs for nitrate (Figure 14) show that this was indeed the case: for the Rhine Plume farm (graph a) high uptake occurred, starting at high nitrate concentrations in winter, and for light levels over 100 µE m$^{-2}$ s$^{-1}$. For the Sound of Kerrera farm (graph b), there was only limited opportunity for high uptake, as under most conditions either light or nutrients were lacking. The Strangford Lough farm (graph c) did not experience high uptake at all. The Norfolk farm (graph c) experienced a good range of conditions that allowed high nitrate uptake. Results for phosphate showed very similar patterns, and are not shown here.

Plotting modelled macroalgae biomass for the Sound of Kerrera farm in a similar way and for the individual years (Figure 15) elucidates the mechanism behind the variability in farm yield in the model (Figure 11f). The final biomass appeared to be correlated not only with the winter nutrient concentration but also with structural biomass in spring, when nutrient concentrations were still elevated and light levels exceeded 50 µE m$^{-2}$ s$^{-1}$ (compare also with the uptake rates, Figure 14b). A sufficient level of initial spring biomass was required to allow for sufficient uptake and storage of nutrients to facilitate the early summer growth. The initial spring structural biomass appeared to be correlated with the combination of light and nutrient concentrations in late autumn/early winter. Hence, it appears that the timing of the onset of increased nutrient levels, and the timing of nutrient drawdown are important determinants of farming success in areas where winter nutrient levels are not elevated and subject to interannual variation.

### 4    Discussion

The modelled production of macroalgae showed a range of responses that may illustrate the actual production that can be expected from commercially operated farms in these locations. Having said this, the model results were not highly accurate for all sites. At Strangford Lough, the modelled winter nutrient concentrations were likely too low, leading to low macroalgae production in the model. This result provides an analogue for potential lack of farming success at sites with

naturally low winter nutrient concentrations.

The model results at the Sound of Kerrera site were realistic, comparing in range with observed winter nutrient concentration levels (higher than at the Strangford Lough site), and also comparing in range with the observed variation in macroalgae production. Modelled production at the Lynn of Lorne site was higher than at Sound of Kerrera, coinciding with higher

modelled winter nutrient concentrations. However, as a side effect of this, macroalgae carbohydrate content was lower.

At the Rhine Plume site, modelled nitrate levels were substantially higher than observed. Despite this, macroalgae production per metre was lower than at Sound of Kerrera and Lynn of Lorne due to less favourable light conditions caused by higher concentrations of suspended solids and the line being deeper below the surface. The modelled macroalgae

contained low concentrations of carbohydrates, as they had continuous access to nutrients. The observed nitrate concentrations at a near-by location suggest limiting conditions in summer, and hence the real farm may yield macroalgae with a higher carbohydrate content.

The Norfolk farm, after compensating for the overestimated modelled suspended particulate matter concentrations by

reducing the depth of the lines below the surface, produced modelled macroalgae biomass per metre of line comparable with the higher values reported for the Sound of Kerrera farm. This production showed good interannual stability, and contained up to 60% carbohydrates. Simulated winter nutrient concentrations were comparable with observed concentrations. There was a slight variation in macroalgae production between the three model grid cells occupied by the farm, in line with a slight gradient in suspended particulate matter concentrations. Even for this farm, which was the largest that was modelled, we did

not find significant changes in temporal averages of other model environmental variables over the period of simulated farming. This is presumably because nutrient requirements of *S. latissima* are very modest, and there is a high level of flushing in the area. Overall, this result supports the potential for this site for macroalgae farming.

The model results suggested that, in areas where winter nutrient levels are modest, farming success could be sensitive to the

timing of the autumn onset and spring draw-down of nutrient levels. This result should be further tested and investigated using more detailed field and laboratory observations.

## 5    Recommendations





This model study did not detect large-scale changes in environmental conditions in the vicinity of the simulated farms. Although this is encouraging, we do not, however, consider this finding to be a generic result, and further and specific investigations should be carried out for specific proposed farm implementations. Such work could include application of and contrasting with other models, and further up-scaling of farm size and intensity to explore safe limits. Moreover, the current

model (as any model) only captured a subset of environmental processes.

The results for the hypothetical Norfolk farm site suggest favourable conditions for commercial macroalgae farming. However, suspended particulate matter concentrations may remain an issue, and accurate regulation of a very shallow depth of line below the surface is probably required. A small-scale field experiment is recommended to test this result in reality.

For the Sound of Kerrera site, with lower nutrient concentrations and variable farm yield, the model suggested a relationship between farm yield and autumn and spring nutrient concentrations coinciding with light at sufficient levels. This suggested relationship should be investigated further, and confirmed with more detailed observations than available for this study, as further understanding of these processes can help to determine minimum required conditions for successful farming.

The model results suggest high rates of macroalgae growth in early summer, accompanied with an increase in carbohydrate content, but also by an increase in mortality. This suggests that there is an optimum window for harvesting, in line with experience from the experimental farms; however, the simulated mortality was not enough to start to reduce biomass. The model suggested differences in this balance between the farm sites, but without further field evidence it is difficult to draw

detailed conclusions. It is recommended to continue the field experiments, and to gather more detailed information on environmental conditions, carbohydrate content and mortality. This could be accompanied by suitable series of shore-based microcosm experiments. Associated modelling work can help to explain and extrapolate such results.

Concerning the model, improvements could be made in the simulation of nutrients and particulate suspended matter. Also,

representations of different farm configurations could be considered (eg. undulating or vertical lines). Other macroalgae species could be included, as well as a capability to model natural, sea-bed attached macroalgae populations. Finally, inclusion of macroalgae grazers in the model could be investigated, as grazing can be a problem.

**Acknowledgements**

This work was carried out under the SeaGas project, led by the Centre for Process Innovation (CPI). The SeaGas project is funded by both Innovate UK (for industrial partners) and BBSRC (for academic partners). The funding scheme is the Industrial Biotechnology Catalyst, grant number 102298. Queens University was supported by BBSRC under grant number BB/M028690/1. The work was carried out under Cefas contract code C6627. The macroalgae modules were developed at NIOZ. Recent elements of the model development were funded by Cefas Seedcorn projects DP261 and DP315.




Development of postprocessing was supported by Defra under Cefas contract FC002. Claire Coughlan, while at JRC (Ispra), created the open-boundary forcing for temperature, salinity and nutrients for the model. ECMWF and BADC are acknowledged for making the atmospheric forcing available. Data from Strangford Lough is from the EnAlgae project, which was funded by the European Regional Development Fund via the INTERREG IVB NWE programme. Data from the

5    Sound of Kerrera farm were funded through the European Commission Seventh Framework Programme (FP7) project - Advanced Textiles for Open Sea Biomass Cultivation (AT~SEA) grant no. 280860.



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





**Table 1. Limiting environmental variables for macroalgae cultivation. Ranges in bold were satisfied within the selected farm area (green rectangle in Figure 2). Between brackets: values suggested by Capuzzo et al. (2014) if different.**

| Variable | Unsuitable | Sub-Optimal | Optimal | Reference |
|---|---|---|---|---|
| Contribution to suitability index | 0 | 0.5 | 1 | |
| Minimum temperature [°C] | <2 | **2-5** (2-4) | >5 (>4) | Bolton and Lüning (1982) |
| Maximum temperature [°C] | Adapted farming methods assumed possible (>18) | Adapted farming methods assumed possible (16-18) | Adapted farming methods assumed possible (<16) | Bolton and Lüning (1982) |
| Wave height [m] | >6 | 4-6 (<1 & 4-6) | **0-4** (1-4) | Buck and Buchholz (2005) |
| Photic depth [m] | <1 (<2) | 1-2 (2-4) | **>2** (>4) | |
| Winter nitrate [mmol m$^{-3}$] | <10 | 10-20 | **>20** | Aldridge *et al.* (2012) |
| Tidal velocity [m s$^{-1}$] | >2 | <0.25 & 1.5-2 | **0.25-1.5** | Buck and Buchholz (2005) |
| Water depth [m] | <4 (<10 & >50) | | **>4** (10-30) | |





**Table 2. Farm description.**

|  | Strangford Lough | Sound of Kerrera | Lynn of Lorne | Rhine Plume | north Norfolk coast |
|---|---|---|---|---|---|
| Latitude | 54.40 | 56.40 | 56.50 | 52.15 | 51.53 |
| Longitude | -5.58 | -5.58 | -5.50 | 4.10 | 0.82-0.98 |
| Line length [m] | 100 | 100 | 100 | 85 | 3500 |
| Number of lines per farm | 21 | 24 | 24 | 1 | 350 |
| Distance between lines [m][a] | 5 | 4 | 4 | - | 50 |
| Depth below surface [m] | 1.0 | 1.5 | 1.5 | 2.0 | 0.3[b] |
| Initial biomass per m line [mg C m$^{-1}$] | 2500 | 2500 | 2500 | 2500 | 2500 |
| Number of plants per m line | 100 | 100 | 100 | 100 | 100 |
| Deployment day of year | 274 | 274 | 274 | 274 | 274 |
| Harvest day of year | 183 | 183 | 183 | 183 | 183 |
| Number of grid cells covered by farm | 1 | 1 | 1 | 1 | 3 |
| Location in Figure 1 | A | B | C | D | E, F, G |

[a]The model worked with an implicit line distance of 1 m.

[b]The depth of the farm for north Norfolk was set to 0.3 m instead of 1.0 m to compensate for the over-estimated SPM

5  concentrations and corresponding lower light levels.

**Table 3. Simulated farm yields at harvest at the end of July ($10^3$ kg C; $10^3$ kg wet biomass between brackets; factor 24.919)**

|  | Farm size (m of line) | 2006/7 | 2007/8 | 2008/9 | 2009/10 | 2010/11 |
|---|---|---|---|---|---|---|
| Strangford Lough | 2100 | 1.1E-1 (2.8) | 1.0E-1 (2.5) | 1.1E-1 (2.7) | 1.0E-1 (2.5) | 8.4E-2 (2.1) |
| Sound of Kerrera | 2400 | 9.7E-1 (2.4E1) | 1.1 (2.8E1) | 7.2E-1 (1.8E1) | 4.3E-1 (1.1E1) | 2.8E-1 (7.0) |
| Lynne of Lorne | 2400 | 2.3 (5.6E1) | 2.5 (6.1E1) | 1.2 (3.0E1) | 8.1E-1 (2.0E1) | 7.8E-1 (1.9E1) |
| Rhine plume | 85 | 6.5E-2 (1.6) | 6.0E-2 (1.5) | 6.0E-2 (1.5) | 5.5E-2 (1.4) | 5.9E-2 (1.5) |
| Norfolk A | 245000 | 2.4E2 (6.0E3) | 2.3E2 (5.8E3) | 2.3E2 (58E3) | 2.5E2 (6.1E3) | 2.5E2 (6.2E3) |





| Norfolk B | 245000 | 2.6E2 (6.4E3) | 2.5E2 (6.2E3) | 2.5E2 (6.3E3) | 2.6E2 (6.6E3) | 2.7E2 (6.7E3) |
| Norfolk C | 245000 | 2.7E2 (6.6E3) | 2.6E2 (6.4E3) | 2.7E2 (6.8E3) | 2.8E2 (6.9E3) | 2.8E2 (7.0E3) |
| Norfolk total | 735000 | 7.6E2 (1.9E4) | 7.4E2 (1.8E4) | 7.8E2 (1.9E4) | 7.8E2 (2.0E4) | 7.8E2 (2.0E4) |





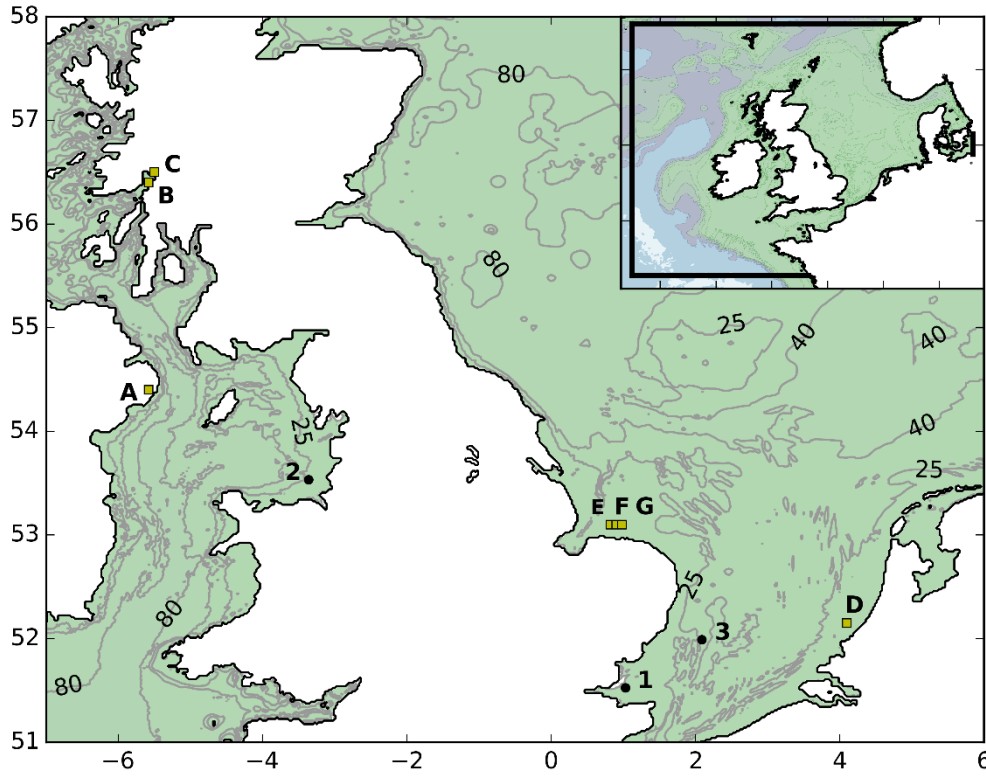

**Figure 1. Study area with SmartBuoy stations (black circles: 1 = Warp Anchorage, 2 = Liverpool Bay, 3 = West Gabbard), and macroalgae farm locations (yellow squares represent the macroalgae farms: A = Strangford Lough; B = Sound of Kerrera; C = Lynn of Lorne; D = Rhine Plume; E-G = north Nolfolk; see Table 2 for more information). Depths are in metres. Inset: north-west**
5 **European shelf seas with model domain boundaries (thick black lines).**







**Figure 2. Potential areas for a commercial farm off the North Norfolk coast. Yellow to brown shading: suitability index. Black: moderately high to high shipping intensity (derived from Marine vessel Automatic Identification System ping data obtained from exactEarth Ltd., http://www.exactearth.com/, for the year of 2013). Lines and hashes: various licensed use (Marine Reference dataset, Defra, collated by the Joint Nature Conservation Committee, 2011). Green rectangle: selected farm area.**



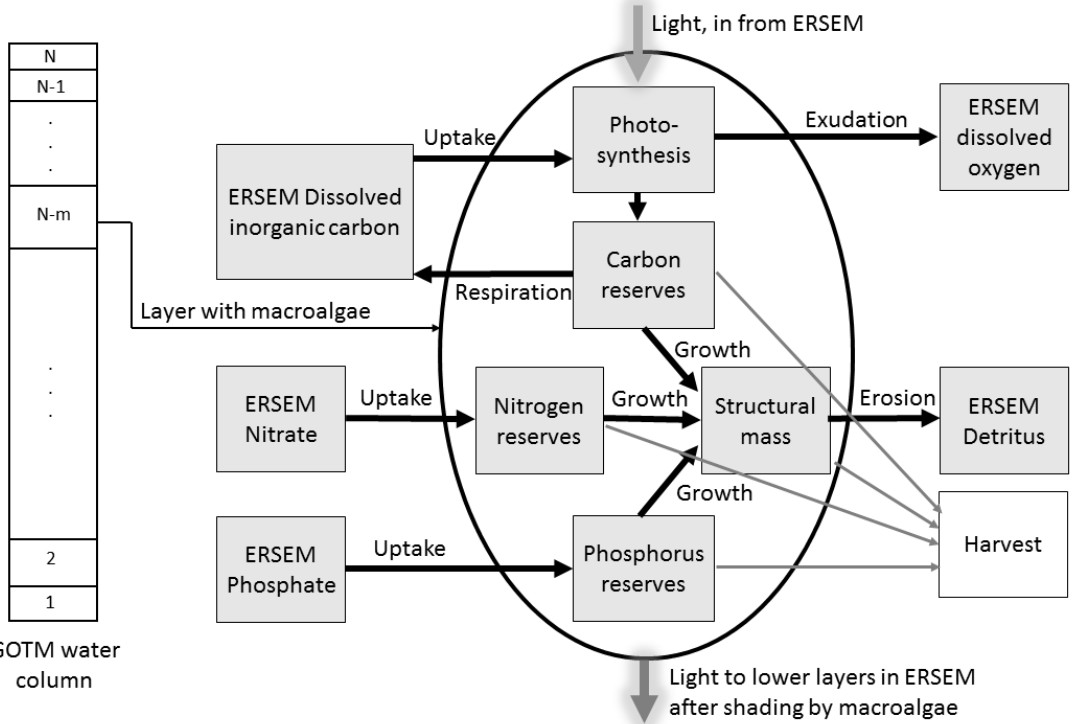

**Figure 3. Schematic representation of the farmed macroalgae in ERSEM-BFM, modified and expanded after Broch and Slagstad (2012).**



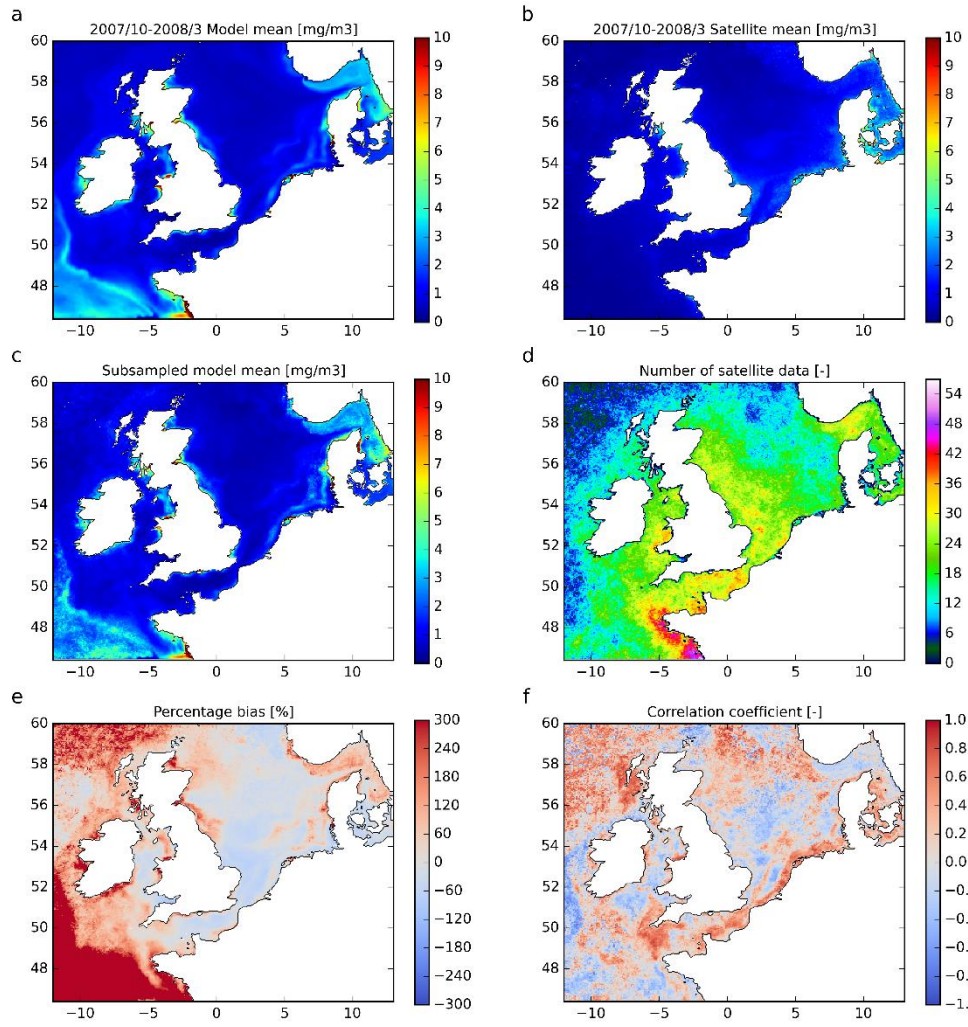

**Figure 4. Comparison of winter chlorophyll-a concentrations between model and satellite, October 2007 to March 2008. a) model mean; b) satellite mean; c) model mean accounting for cloudy days; d) number of clear days from satellite; e) relative model bias; f) correlation coefficient.**





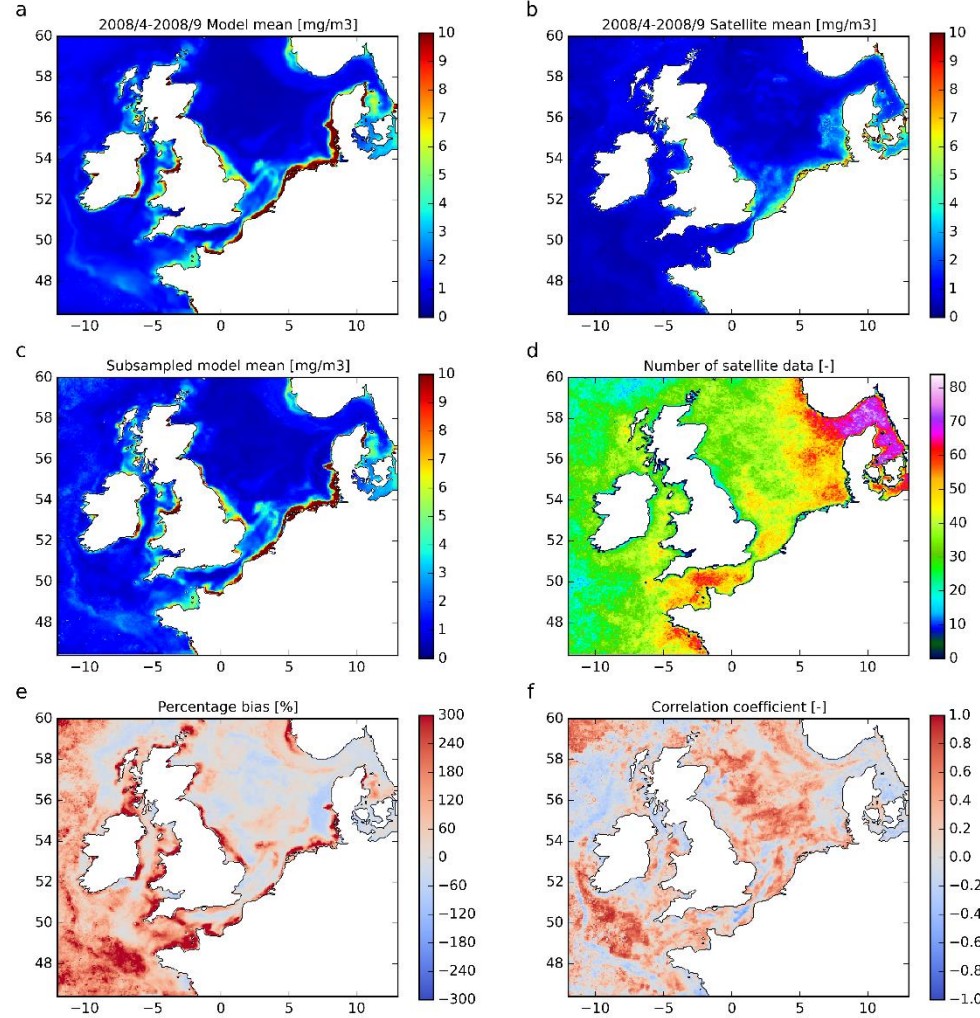

**Figure 5. Comparison of summer chlorophyll-a concentrations between model and satellite, April 2008 to September 2008. a) model mean; b) satellite mean; c) model mean accounting for cloudy days; d) number of clear days from satellite; e) relative model bias; f) correlation coefficient.**



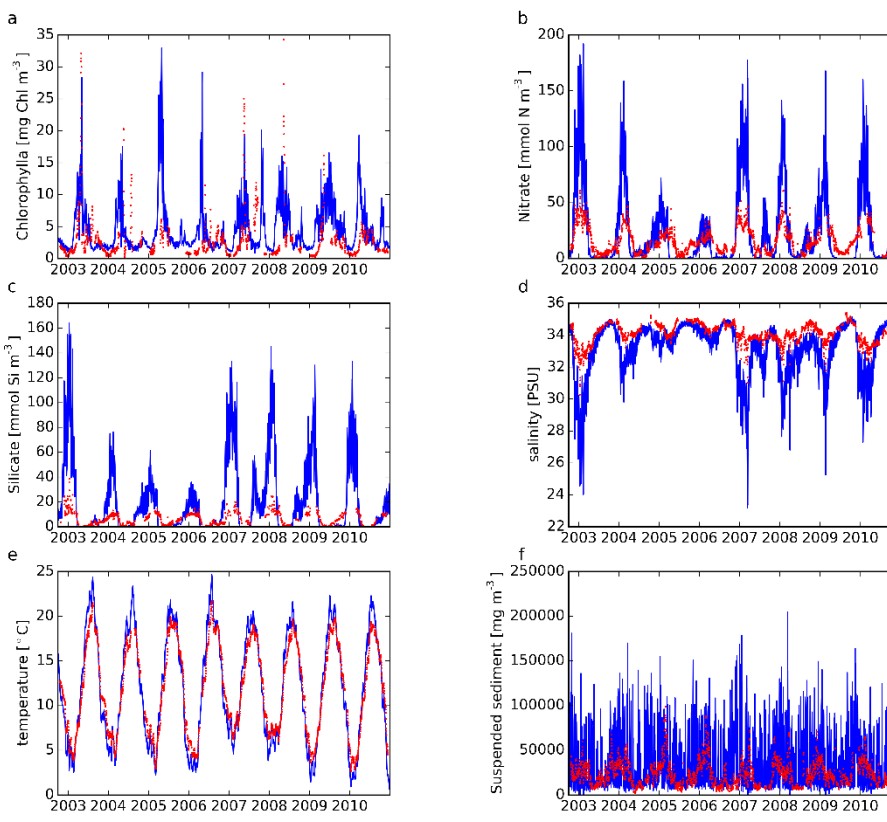

**Figure 6. Time-series comparison with Warp Anchorage SmartBuoy, surface. Blue: model, red: observations. a) Chlorophyll-a concentration; b) nitrate concentration; c) silicate concentration; d) salinity; e) temperature; f) suspended sediment concentration.**





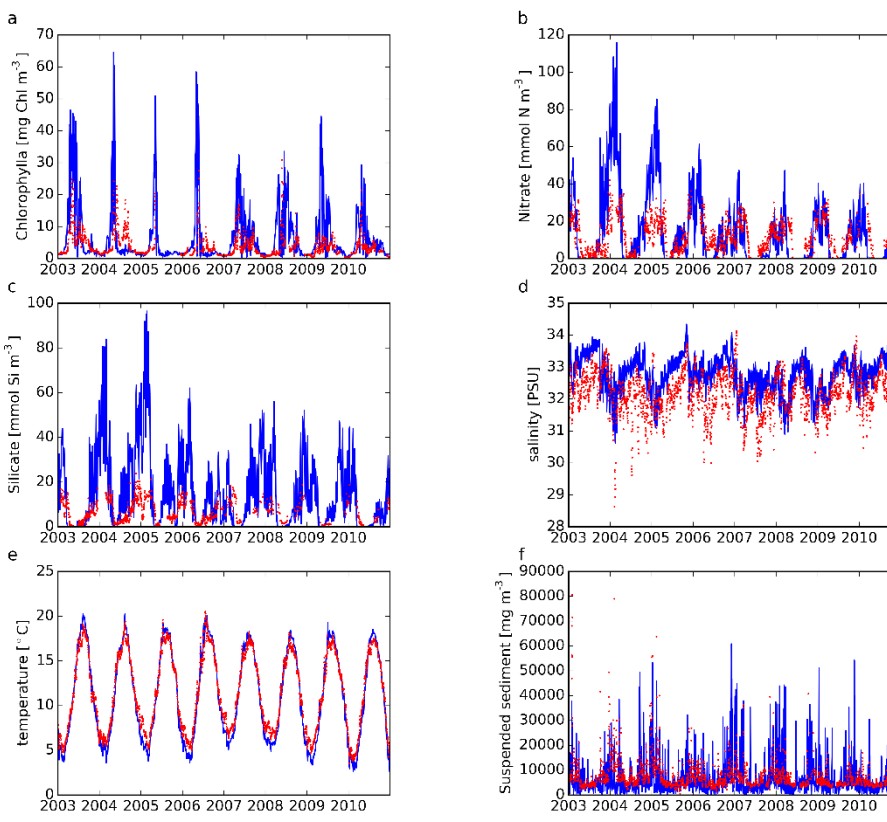

**Figure 7. Time-series comparison with Liverpool Bay SmartBuoy, surface. Blue: model, red: observations. a) Chlorophyll-a concentration; b) nitrate concentration; c) silicate concentration; d) salinity; e) temperature; f) suspended sediment concentration.**



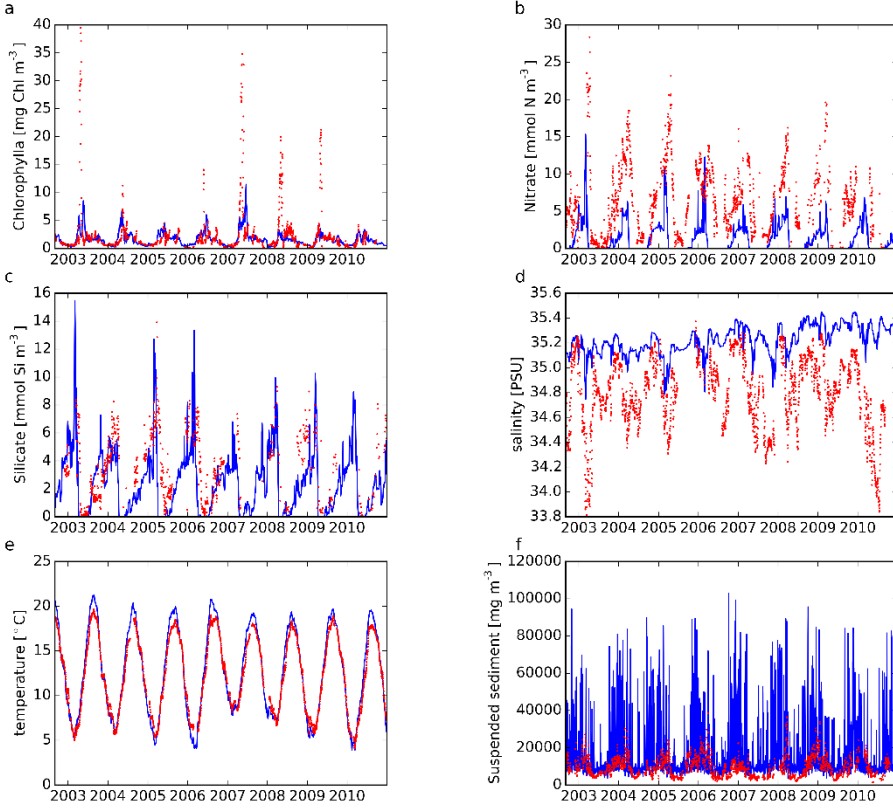

**Figure 8. Time-series comparison with West Gabbard SmartBuoy, surface. Blue: model, red: observations. a) Chlorophyll-a concentration; b) nitrate concentration; c) silicate concentration; d) salinity; e) temperature; f) suspended sediment concentration.**






**Figure 9. Model results for the Strangford Lough farm site. a) surface nitrate concentration; b) surface phosphate concentration; c) total extinction coefficient at the surface (excluding contribution by macroalgae); d) irradiance at the surface; e) surface water temperature; f) macroalgae carbon biomass (structure + carbohydrates) per m of line; g) mass of macroalgae structure over total (structure + carbohydrates) macroalgae mass ratio; h) relative mortality of macroalgae structure; i) C/N ratio of macroalgae; j) C/P ratio of macroalgae; k) farm nitrogen uptake; l) farm phosphate uptake. Black dots in f) are observations from the 2012-2013 deployment.**







**Figure 10. Model results for the Sound of Kerrera farm site. a) surface nitrate concentration; b) surface phosphate concentration; c) total extinction coefficient at the surface (excluding contribution by macroalgae); d) irradiance at the surface; e) surface water temperature; f) macroalgae carbon biomass per m of line; g) mass of macroalgae structure over total macroalgae mass ratio; h) relative mortality of macroalgae structure; i) C/N ratio of macroalgae; j) C/P ratio of macroalgae; k) farm nitrogen uptake; l) farm phospate uptake. Black dots are observations: in a) and b) from nutrient samples, in d) and e) monthly averages from a data logger.**







**Figure 11. Model results for the Lynn of Lorne farm site. a) surface nitrate concentration; b) surface phosphate concentration; c) total extinction coefficient at the surface (excluding contribution by macroalgae); d) irradiance at the surface; e) surface water temperature; f) macroalgae carbon biomass per m of line; g) mass of macroalgae structure over total macroalgae mass ratio; h) relative mortality of macroalgae structure; i) C/N ratio of macroalgae; j) C/P ratio of macroalgae; k) farm nitrogen uptake; l) farm phospate uptake.**






**Figure 12. Model results for the Rhine plume farm site. a) surface nitrate concentration; b) surface phosphate concentration; c) total extinction coefficient at the surface (excluding contribution by macroalgae); d) irradiance at the surface; e) surface water temperature; f) macroalgae carbon biomass per m of line; g) mass of macroalgae structure over total macroalgae mass ratio; h) relative mortality of macroalgae structure; i) C/N ratio of macroalgae; j) C/P ratio of macroalgae; k) farm nitrogen uptake; l) farm phospate uptake. Black dots are observations from the near-by Noordwijk transect at 10 km offshore collected by RIKZ (live.waterbase.nl).**





**Figure 13. Model results for the western-most grid cell of the north Norfolk farm site. a) surface nitrate concentration; b) surface phosphate concentration; c) total extinction coefficient at the surface (excluding contribution by macroalgae); d) irradiance at the surface; e) surface water temperature; f) macroalgae carbon biomass per m of line; g) mass of macroalgae structure over total macroalgae mass ratio; h) relative mortality of macroalgae structure; i) C/N ratio of macroalgae; j) C/P ratio of macroalgae; k) farm nitrogen uptake; l) farm phospate uptake. Black dots in c) are the $k_d$ contribution by SPM, calculated from in-situ SPM samples collected in the years 1996-2000 using the relationship derived by Devlin et al. (2009), and projected onto 2006-2007.**



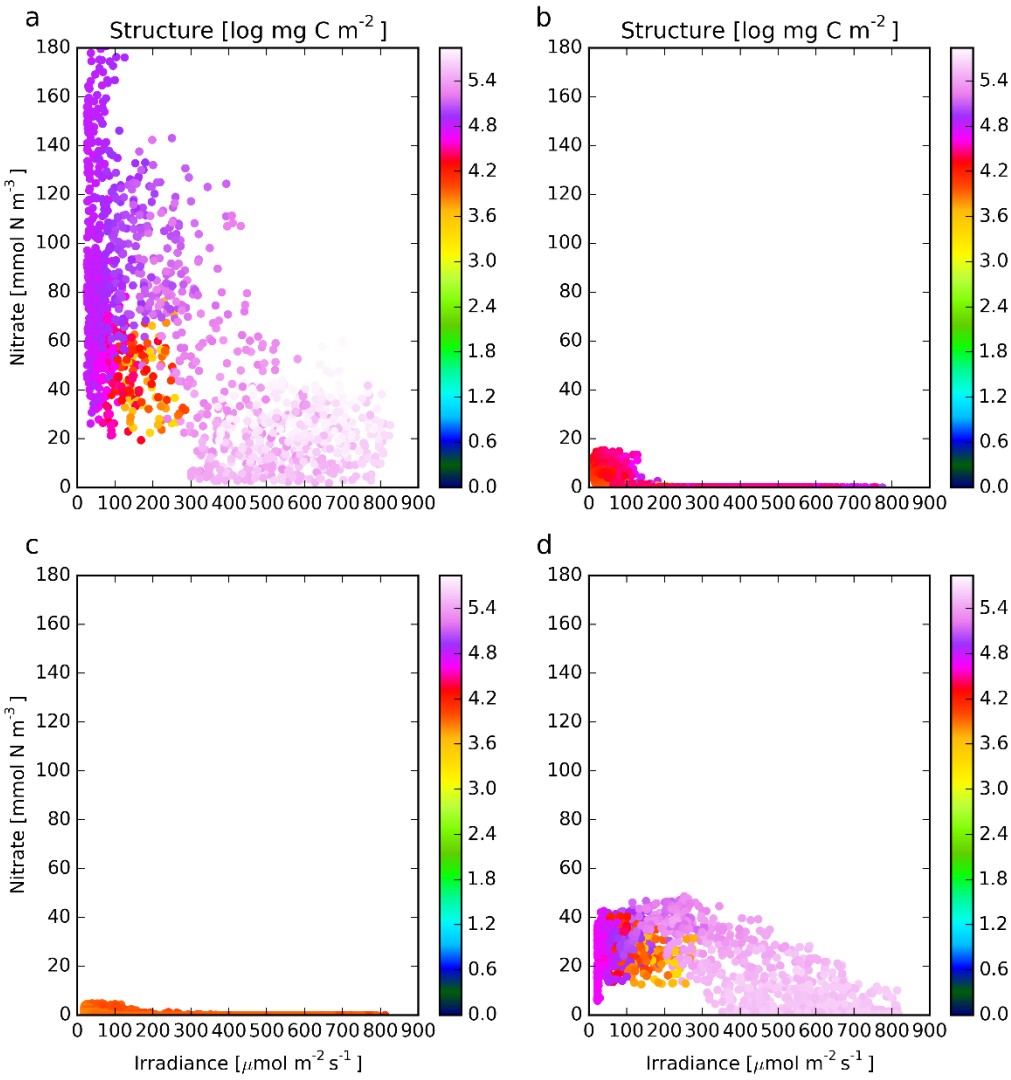

**Figure 14. Logarithm of structural biomass as a function of irradiance and nitrate concentrations in the model. a) Rhine Plume farm, b) Sound of Kerrera farm, c) Strangford Lough farm, d) Norfolk farm.**



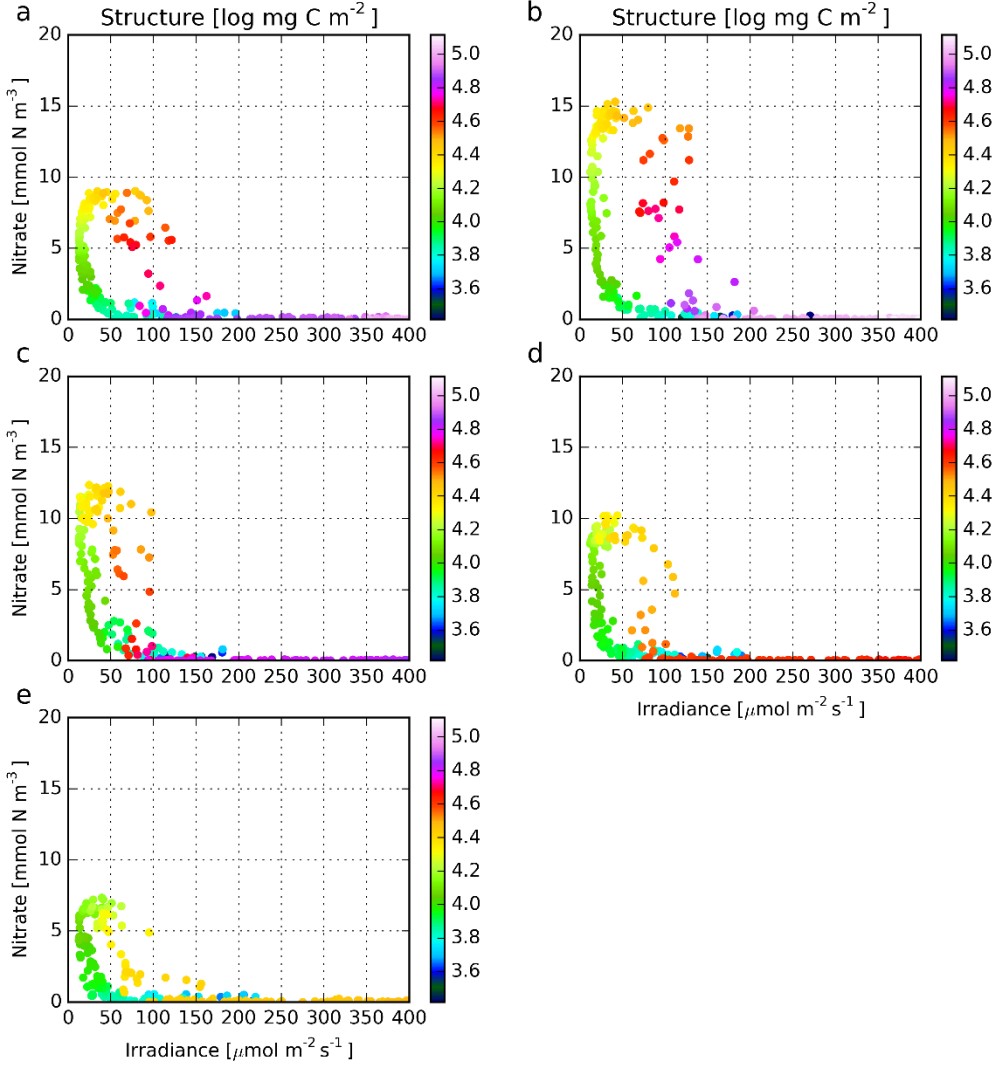

**Figure 15. Logarithm of structural biomass as a function of irradiance and nitrate concentrations in the model for the Sound of Kerrera farm. a) 2006, b) 2007, c) 2008, d) 2009, e) 2010.**