# Peer review of "Modelling potential production of macroalgae farms in UK and Dutch coastal waters"

_Biogeosciences, 2017_

## Referee Comment (RC1) · Anonymous Referee #1 · 14 Jul 2017

The manuscript with the title "Modelling potential production and environmental effects of macroalgae in UK and Dutch coastal waters" by Johan van der Molen et al. provides estimates for existing and hypothetical seaweed farms within the UK and the Netherlands based on the simulations with a 3D marine ecosystem model. For the production site the model does not only provide estimates for the overall production but also on the quality in terms of carbon content of the macroalgae. In contrast, the summary of the environmental effects was that they were not detectable when comparing the reference run and the scenario run. For my understanding the later conclusion is too simple to justify the expression "environmental effect" in the title. The authors should re-think if they can support this claim in the title, some hints will be expressed in detail below, or

[Figure]

drop this part in the title. Since clarification are needed on the production part as well, the manuscript should be published after major revision.

My suggestion to back up the claim to provide information related to "environmental effect" would be to calculate the flux of nutrient and carbon uptake by the farmed macroalgae in comparison to the phytoplankton usually used in the model. The nitrogen and phosphate uptake is already presented as time series for each farm site in Fig. 9 – 12 in the graphs k and l. With this quantitative information one could underpin in which way the phytoplankton, in the grid call where the aquaculture is applied, was still able to develop in a way that is not detectable by simply plotting concentration differences. Especially at sites where the nitrogen concentration is depleted is summer it would be very interesting to see how the phytoplankton could manage its uptake in comparison to the newly introduced macroalgae. In addition this information could be supported by the rather simple calculation of the nutrient content in the water column exposed to farming both in winter and in summer, or other crucial important period like autumn. In doing so one can still not define in which way changing transport of either nutrients or phytoplankton led to the indistinguishable concentration differences between the two runs, but one would provide some quantitative background information on the "environmental effects".

One more small detail, in chapt. 3.2 "Environmental effects" the "differences between the two reference runs" (Page 12, line 7) should rather be the difference between the reference run and the scenario with the seaweed farm application or is this a misinterpretation?

On the other hand the study goes beyond a simple feasibility study by providing information not only on the potential harvest that can be expected but also about the seaweed quality in relation to the use for biofuel in terms of the carbohydrate content. Especially for this "fledging industry" the relation of nutrient limitation leading to higher carbon content in context with higher product quality should be expanded into more general consideration that go beyond the individual site description. With Hori-

zon 2020 calls on the co-use of technical structures like offshore windfarms in the North Sea for aquaculture or seaweed farms, this aspect has the potential to go beyond a pure coastal application and would raise the impact of this study towards more general consideration in this context. These additional consideration would compensate the problem that the model over- or underestimates the individual nutrients and/or chlorophyll-a concentrations at different validation or farm sites, so that it is difficult to draw more general conclusions from these aspects of the model performance.

When describing the nutrient validation results for Sound of Kerrera farm site (Fig. 10a and b), the question is not only on the nutrient concentration reached in comparison to the measurements. The more profound question is why the measurement show a rather different cycle compared to the simulated concentration. Is there any local source that is not reproduced in the model that brings about these differences?

Since the mortality term is one of the key parameter for the seaweed yield a more detailed description of this process is needed. The simple hint toward "erosion" does not help the reader in which way the mortality process interacts within the simulation throughout the year. As a simple example, when at site A (Strangford Lough) the macroalgae biomass is about an order of magnitude lower than the observed one but the mortality is low throughout the farming cycles, one would interpret that the light and/or nutrient condition were not suitable in the model to efficiently reproduce the measured biomass. This implies that the mortality pressure is not an important factor when it is applied as proportional to the biomass, but this is not clear from the incomplete explanation provided.

Smaler details: For a more simple way of attributing the different sites, the characterisation A-G as used in Fig. 1 should be used as site specification next to the full locality description of the farms throughout the paper. In this context the site specification for each farm and validation location should be highlighted in a different colour in Fig. 1 rather than simply black. In addition, since the Doggerbank and the Norwegian Trench are used in the characterisation of the North Sea hydrodynamics, these features should

be more pronounced in the topography map in Fig. 1, e.g. in the smaller map showing the full domain.

Is Fig. 2 in this full detailed information overview really needed?

The general statement on the model confirmation that "These results are not reproduced here" (page 10, line 27) should be placed on top of this subchapter.

Page 20 line 19 The first author is Jo Folden not Foden

―――――――――――――――――――――

---

## Referee Comment (RC2) · F Große (Referee) · 21 Jul 2017

**General comments**

The manuscript by van der Molen et al. deals with the representation of macroalgae – more specifically kelp – farms in the physical-biogeochemical model system GETM/ERSEM-BFM. Their study aims for the model-based analysis of their impact on the ecosystem as well as the productivity of such farms. A number of near-shore (four existing and one non-existing, but potentially suitable) farm sites off the British and Dutch coasts are used for this analysis.

The increasing interest in macroalgae farms, which is well described in Sect. 1.1,

provides a good background for this objective. Due to model limitations, e.g., with respect to spatial resolution or agreement with SPM or chlorophyll a observations, the authors clearly state that their study rather constitutes a "proof of concept" than a detailed analysis of the likely environmental impact and productivity of these farms (at end of Sect. 1.1).

Despite this clearly stated limitation, I consider the objective of the study as relevant and feasible for publication. However, I have a few major and some minor points that should be addressed by the authors.

My first major point of criticism is that the "environmental effects" included in the title are barely addressed in the manuscript (except for the information, that there are basically none in Sect. 3.2) and, therefore, either the title should be adapted or the manuscript should be extended with respect to that. For the relevance of the manuscript, I would propose to do the latter and will provide some suggestions in my specific comments below.

Second, a partial re-ordering of the Sects. 1 and 2 (and subsections), from my perspective, would clearly improve the readability of these parts of the manuscript.

Third, I would request a more detailed discussion of the model setup and the implementation of the kelp farms (and a more detailed description of the latter). For the latter, this could be especially relevant in relation to the potential environmental effects and may provide the reader with a better understanding of why the effects appear to be negligible in the present study.

Provided this more detailed description and discussion of the limitations and constraints, this study may serve as a first guideline for investigating environmental effects and performance of macroalgea farms using 3D physical-biogeochemical models. Therefore, I recommend reconsidering the manuscript for publication after major revision.

**Specific comments**

1. Most parts of the last paragraph of Sect. 1.1 (page 3, end of line 16 to line 21)

rather sound like a discussion/conclusion to me and I would suggest moving these parts correspondingly. Instead the authors could complete the short outline of the manuscript in this paragraph.

In this paragraph the authors also mention that the large-scale model allowed for the inclusion of all farms in one model (page 3, line 16). The phrasing makes it sound beneficial – which is indeed the case from a technical point of view (setting up one model instead of one for each study site). However, considering the related limitations, this should be discussed critically in the discussion section.

2. Organisation of Sects. 1.2 to 2.2.2: I had difficulties going through this part of the manuscript as it partly caused a back and forth reading. This mainly relates to the fact that the characteristics of the study sites (bathymetry, hydrography etc.) are provided in Sect. 1.2, followed by the description of the considered kelp species (*S. latissima*) in Sect. 1.3, again followed by a now more technical description of the setup of and – if existent – the sampling at the different farm sites in Sects. 2.1.

I see the difficulties with the separation/combination of Sects. 1.2 and 2.1 as the former is a more general description, while the latter is more of a methodological nature. However, considering the analysis of selected study sites as part of the methodological approach, I would propose to combine the currently two subsections on the individual study/farm sites into one subsection for each farm. These subsections should then also include the representation of the individual farms in the applied model setup (e.g., position in the vertical – are all farms in the surface layer of the model?). These (general and technical) descriptions of the study sites would then be part of the methodology section (Sect. 2), preferably after the description of the implementation of the macroalgae farms to ERSEM (Sect. 2.2.2). Current Sect. 2.2.3 would then become Sect. 2.2.4.

In relation to this suggestion, current Sect. 1.3 (description of *S. latissima*) would become Sect. 1.2. From my perspective, this is feasible as Sect. 1.1 already indicates that this study focuses on UK and Dutch sites, where *S. latissima* is a species of

interest.

3. Sect. 1.2.1 (Southern North Sea): As the North Sea study sites are both located in the south-western North Sea and kelp farms will most likely be placed in the shallower southern North Sea and coastal areas (I assume, correct me if I am wrong), I would shorten the North Sea description to the parts relevant for this region, and rather extend those parts slightly.

For instance, the Norwegian Trench and Skagerrak (page 3, lines 26-27) and the stratification characteristics in the central and northern North Sea (page 4, line 4) are less relevant for the context of this study, whereas the paragraph on North Sea primary production (page 4, lines 12-18) could be underpinned by some literature and more focused on the southern/coastal North Sea (those regions suitable for macroalgae farms).

Differences in productivity between the Norfolk and Rhine plume sites may be stated in the subsequent paragraph (page 4, lines 20-25).

4. Sect. 2.1.1. (Strangford Lough): I am no specialist in macroalgae at all, which is definitely one reason for the following two questions:

The actual farm consists of 2 lines with *S. latissima* (that is also implemented to ERSEM) and 19 lines with other species. Though, for the model 21 lines with the former are assumed. To what extent may this affect the results? Are there strong inter-species differences, e.g., with respect to nutrient requirements? Maybe a short note on that can be made in the discussion.

It is assumed "that the dry plant material consists predominantly of CH2O groups" (page 6, line 31) – is this a reasonable assumption? (Maybe underpin with literature.)

5. Sect. 2.1.2 (Sound of Kerrera . . .): The last sentences (page 7, end of line 10 to line 14) should be moved to the corresponding time series results or even to the discussion.

6. Sect. 2.2.2 (Macroalgae farms in ERSEM): It is stated that the farms are implemented to the model by means of number of lines, line length etc. for which the parameter values are provided in Table 2. However, no information is provided on how the actual description/implementation in the model is done, nor is a reference provided containing such description (if existent). From my point of view this step is quite essential when presenting the study as a "proof of concept". As I could not find any other literature attempting to include macroalgae farms in a large-scale 3D model, I assume that this manuscript presents the first approach. In this case the technical description of the implementation needs to be part of the publication – not necessarily as part of the main text, but in an (electronic?) appendix or as supplementary material. Regarding the model schema (Fig. 3) I also wonder whether the light climate in all parts of the water column below the farms is affected, i.e., including the below-farm parts of the grid cell with the farm itself, or only the grid cells below the farm grid cell. Depending on the vertical extent and the position of the farm, this may influence surface layer primary production by other algae (e.g., diatoms)

7. Sect. 2.2.3 (Model scenarios): For the farm scenarios, the model was run from 1 October to end of July of the subsequent year (page 9, line 33). Again a question as a non-specialist: How does this relate to the actual farming practice?

8. Sect. 3.1 (Model confirmation): Although it is probably the case, it would help to clearly state at the beginning of this section that the provided model confirmation/validation is based on the reference run.
With respect to the satellite vs. model comparison (Figs. 4 and 5) I wonder whether the map section could be reduced, more focusing on the regions of interest of this study (e.g., similar to Fig. 1). This would also allow for a more detailed description/discussion of the model quality in the areas of interest. Furthermore, it should be mentioned which months are used for the "summer", respectively, "winter" maps.

With respect to the time series (Figs. 6-8), I have several comments/questions:
On page 11, lines 15/16, it is stated that peak spring bloom chlorophyll a at Warp
Anchorage (Fig. 6) is about 10 mg m-3. Does this refer to the observations as the
model shows much higher values (up to 30 mg m-3)? If so, this should be made clear.
Since chlorophyll is the main quantity used in the validation, it would help to provide
brief information on how the chlorophyll concentration is calculated/derived (prognos-
tic, diagnostic using fixed/variable chlorophyll-to-carbon ratios) by the model in the
model description (a reference is sufficient).
I further wonder about the large discrepancy between simulated and observed salinity
and in relation to that – as also mentioned by the authors – nutrients. Does this relate
to the applied river forcing? Or are there other likely causes? Although not in the focus
of the study, a brief comment on that would be useful.

9. Sect. 3.2 (Environmental effects): As stated in my general comments this
aspect is barely addressed in the manuscript. The authors mention "maps of differ-
ences in biogeochemistry and plankton dynamics" without further specifying what
kind of maps. Considering, e.g., that the kelp farms affect the light availability in
the deeper model layers (as indicated in the schema in Fig. 3), I wonder whether
only surface maps were analysed or whether quantities in deeper layers (affected by
potential changes in the light climate) were also analysed? At least at the farm scale, I
would expect changes in the light climate in the water column below each farm. Such
change may not necessarily lead to distinct changes in the biogeochemistry, due to
the small farm sizes or nutrient limitation, however, it may be used as an indicator
when considering an up-scaling, i.e., larger farms. Therefore, I would propose to either
specify what kind of quantities were analysed without including additional results,
or to show and briefly discuss the difference map of one meaningful quantity (e.g.,
light availability in the deeper layers during the phytoplankton spring bloom period
and/or corresponding spring bloom primary production). This would strengthen the
manuscript with respect to that objective of the study.

Furthermore, the authors refer to "differences between the two reference runs" (page 12, line 7) – based on the scenario description (Sect. 2.3) I do not understand what the second reference run is? I suppose the scenario run (incl. the farms) is referred to? This should be made clear.

10. Sect. 3.3 (Strangford Lough time series): A short in-text definition of the structure-to-mass ratio (as given in the caption of Fig. 9) would be helpful. Principally, the authors may consider defining a Sect. 3.3 "Kelp farm performance/productivity" (or similar) with the current Sects. 3.3-3.8 as subsections (then 3.3.1-3.3.6), providing a more distinct separation from the "environmental effects".

11. Figs. 9-13: panels k and l: What do negative uptake rates mean? Do the plots show the net uptake, i.e., uptake minus respiration? This should be clarified in the text. Panels c and d: To me it is not fully clear to what the provided extinction coefficients relate – is it the one in the water column above the kelp farms or the average of the grid cell in which the farms are located? I suppose it is the former as the figure captions state "excluding contribution of macroalgae". However, it would help clarifying this in-text.
Similarly, I wonder about the irradiance – is it the irradiance at depth of the macroalgae, that directly at the sea surface, or that at the centre of the grid cell? Considering, e.g., the Rhine plume farm, with a line depth of 2m and high extinction coefficients, this may result in well different values.
It may help to include a brief general description of the quantities displayed in the time series at the beginning of the time series section.

12. Discussion/Recommendations: In relation to the authors' statement early in the introduction ("this study is a proof of concept") and the general performance/setup of the model (e.g., partly unsatisfactory reproduction of observed chlorophyll concentrations, coarse spatial model resolution), I understand that those two sections are

rather general and focus on the discussion of the results on the farm performance, and provide suggestions for improved analyses of environmental impacts and farm performance.

However, I would request a more detailed discussion of which limitations of the study affect the results in what way (e.g., low spatial resolution vs. small-scale environmental effects or small farm size vs. larger scale environmental effects). Part of this is already indicated during the course of the manuscript (e.g., small farm size in Sect. 3.2). Following the "proof of concept" approach of the authors, a discussion subsection dedicated to the limitations of the setup in relation to the model outcome should be provided, incl. potential effects of an improved setup, where applicable. Some of the potential limitations I raised in the previous comments.

Such subsection would also provide a good basis for the recommendations section, in which the suggestions for an improved study setup can be made. From my perspective, setting up the discussion/recommendation like this would clearly strengthen the "proof of concept" aim and provide a good basis for future, more detailed studies taking into account the suggestions by the authors.

**Technical corrections/comments**

1. The authors should go through the in-text citations thoroughly and check for consistency regarding punctuation (e.g., commas before years, semi-colons between multiple citations), ordering of multiple citations (chronological or alphabetical – if I am not mistaken the journal has a preference for one of the two), in-text author names and names in the reference list (e.g., "Grosse" in-text, "Große" in the references; in "van der Molen" the V is partly uppercase, partly lowercase).

2. When providing ranges of a quantity (e.g., 15-25m) or areal extents (e.g., 5x5km) I would recommend providing the unit after both numbers.

3. Abstract: Line 2: I would rather write "and for biofuel production".

4. Introduction:

Page 2, lines 26/27 and 29/30: "associated with high biodiversity is doubled.

Page 4, line 13: "reduced" instead of "reducing"?

Page 4, line 15: "matter" instead of "material"

Page 4, line 31 to page 5, line 2: I would propose re-ordering these sentences such that there is no jump in the description from currents to depth and back to currents.

Page 5, line 4: the farm site "is" located (instead of "was")

Page 5, line 10: "The site range from 15-25m depth" sounds a bit odd. Maybe "The depth ranges from 15-25m at the site"?

Page 5, line 27: Would it be suitable to write "nitrate" instead of "nitrogen" as only nitrate uptake by kelps is considered?

Page 5, lines 23-30: I would change the order of the two paragraphs, as the latter is related to kelp in general while the former is region-specific.

5. Methods

Page 6, lines 25 and 30: "MPA" is used without introducing the abbreviation (line 25), while later "Marine Protected Area" is used (line 30).

Page 6, last paragraph: I would suggest to include a sentence stating that the conversion factor used in Table 3 (24.919) results from the combination of the two factors listed in this paragraph.

Page 7, line 17: The abbreviation "ROFI" is introduced on page 3, line 11, so simply use ROFI here.

Page 7, line 29: I think, 2 digits after the comma are sufficient for the geographical information of the site.

Page 8, Sect. 2.2.1: Some links are shown as hyperlinks, others are not. Should be consistent.

Page 9, line 15: typo in "diynamics"

Page 10, lines 16/17: Should the parenthesis be closed after the link, and the closing

parenthesis at the end of the sentence be removed?

Figure 2 has a rather poor quality/low resolution and the green box indicating the Norfolk kelp farm is hard to find as the map contains a lot of information. I wonder whether this map is actually necessary or if the in-text description is sufficient. If the authors prefer to keep this figure, a smaller map section may help – or addition of an arrow pointing to the farm site. In case of keeping the figure, its resolution needs to be increased.

6. Results

Page 12, line 21: I think it should be 0.05 kg C m-1 instead of 0.5 g C m-1.

Page 12, line 24: typo in "carbohydratate"

Page 13, line 13: It is stated that Lynn of Lorne shows a slightly higher yield than Sound of Kerrera. There's a factor of 2 between most of the years that is clearly more than "slightly higher".

Page 13, very last sentence (ending page 14): should be moved to the description of the study site and its representation in the model in the methodology section.

Page 14, line 3: "Mortality did not increase as much as at some other sites". To my understanding this only applies to the Lyne of Lorne farm among "the other sites".

Page 14, Sect. 3.7 and Table 3: As the in-text description only refers to the wet biomass, I would display these numbers first in the table and show the harvest in parentheses or even omit the harvest and only state in the table caption that wet biomass was calculated from the harvest with reference to updated methods section (see my previous comment on the conversion factor).

I would further omit the information on the individual Norfolk farm grid cells in the table, as the differences are quite small. Related to that the hint on the differences in the discussion could be removed (page 15, lines 22-24).

Figs. 9-13: Panels d: The unit of the irradiance is $\mu$mol m-2 s-1, however, in Sect. 3.8 (page 14, lines 14 and 23) $\mu$E m-2 s-1 is used. Although both are in fact the same, consistent usage of one of the two units (preferably the latter) might be helpful for the

reader not too familiar with this irradiance unit.

Panels k: Those could be named as nitrate uptake (as the parameterisation by Bloch and Slagstad only considers nitrate uptake).

Fig. 14: I would recommend re-ordering the figure panels according to the order in which the study sites are described in the text and presented in Figs. 9-13 (excl. Lyne of Lorne)

7. Discussion

Page 15, lines 12-14: It is stated that kelp production at Rhine plume was lower than in the Sound of Kerrera and Lyne of Lorne. For the former, this is obviously not the case (see Figs. 10f and 12f). Also, light availability and extinction are quite similar for Rhine and Sound of Kerrera sites.

Page 15, lines 19-21: There is a factor of 2 between the production per metre between Norfolk and Sound of Kerrera, which I would not describe as "comparable".

Page 16, line 2: "although" in combination with "however" sounds like a doubling to me. Maybe omit the latter?

Page 16, line 25: a period is missing in "eg."

---

## Referee Comment (RC3) · Anonymous Referee #3 · 8 Aug 2017

I am the 3rd reviewer of this manuscript and, having read the other reviews, I agree with most of the comments made by the reviewers so I am not going to repeat those, although I will highlight specific ones I find more relevant, and I will concentrate on a general evaluation.

Overall, I found the article interesting and eventually deserving publication. The article is well written but I agree with Fabian Grosse's comments 1 and 2 about structuring and organisation. I have a number of general comments, below, that I think should be considered before the manuscript is acceptable for publication.

It has been highlighted by both the authors and the reviewers that this is largely a "proof

of concept" study and I agree with this. In that sense, I am not particularly surprised that the environmental effects of the macroalgae farms were found to be negligible, taking into account the experimental and small scale nature of all of these farms, with the exception of the "hypothetical" one off the Norfolk coast. Incidentally, the suggestion by Referee #1 of using a "comparison to the phytoplankton usually used in the model" may be an interesting way of putting their environmental effect into context. Another consideration is that ERSEM does not seem to do such a good job of matching observational data. This does not surprise me because I fully acknowledge the difficulties involved (e.g. due to lack of a sufficiently comprehensive forcing dataset, etc.) but it does detract somewhat from the "real-life" applicability of the present study. Therefore, I would be tempted to suggest a lesser emphasis on how the model replicates observational data (if you have those data, by all means present the comparisons but maybe just present a subset of these in the main body of the text and move the rest to supplementary material) and to dedicate more space to: 1) the technical description of the new aspects of the model (the implementation of macroalgae farms, as suggested by Fabian Grosse's comment no. 6), probably by a combination of providing a bit more detail in the main text, in addition to supplementary material; and 2) the potential large-scale development of macroalgae farming and its environmental effects. In that sense, I found the combination of a number of "real" small scale farms and a large "hypothetical" one a bit unsatisfactory; maybe the real farms should be used to illustrate how well the model fits observations (but taking into account my comments above) and a number (>1) of larger ("commercial") scale hypothetical farms could be used on a separate simulation exercise to illustrate the harvest potential (in terms of quantity and quality) of a macroalgae farming industry. This relates to the comment made by Referee #1 about "more general consideration that go beyond the individual site description". This may also help address the issue of potential environmental effects, which in the present version (as identified by both reviewers) receives insufficient attention to deserve an explicit reference in the title. Finally, the model limitations (Fabian Grosse's comment no. 12) should also be discussed, in particular scale and resolution aspects, as well as

what aspects of the model should be developed and how.

---

## Author Comment (AC1) · 19 Sep 2017

The manuscript with the title "Modelling potential production and environmental effects of macroalgae in UK and Dutch coastal waters" by Johan van der Molen et al. provides estimates for existing and hypothetical seaweed farms within the UK and the Netherlands based on the simulations with a 3D marine ecosystem model. For the production sitethemodeldoesnotonlyprovideestimatesfortheoverallproductionbutalsoonthe quality in terms of carbon content of the macroalgae. In contrast, the summary of the environmentaleffectswasthattheywerenotdetectablewhencomparingthereference run and the scenario run. For my understanding the later conclusion is too simple to justify the expression "environmental effect" in the title. The authors should re-think if they can support this claim in the title, some hints will be expressed in detail below, or drop this part in the title. Since clarification are needed on the production part as well, the manuscript should be published after major revision.

My suggestion to back up the claim to provide information related to "environmental effect" would be to calculate the flux of nutrient and carbon uptake by the farmed macroalgae in comparison to the phytoplankton usually used in the model. The nitrogen and phosphate uptake is already presented as time series for each farm site in Fig. 9 – 12 in the graphs k and l. With this quantitative information one could underpin in which way the phytoplankton, in the grid call where the aquaculture is applied, was still able to develop in a way that is not detectable by simply plotting concentration differences. Especially at sites where the nitrogen concentration is depleted is summer it would be very interesting to see how the phytoplankton could manage its uptake in comparison to the newly introduced macroalgae. In addition this information could be supported by the rather simple calculation of the nutrient content in the water column exposed to farming both in winter and in summer, or other crucial important period like autumn. In doing so one can still not define in which way changing transport of either nutrients or phytoplankton led to the indistinguishable concentration differences between the two runs, but one would provide some quantitative background information on the "environmental effects".

*In principle, comparing the nutrient uptake with that of phytoplankton to get more information on environmental effects is a good idea. However, from a practical point of view this is not possible with the current simulations as nutrient uptake by phytoplankton was not stored, and we are not in a position to repeat the model runs. From a biological point of view, it is not immediately clear what the meaning of such a comparison would be, because i) phytoplankton is advected by the currents, hence spends only a limited amount of time within the farm area and may compensate when it arrives elsewhere; within the farm area the phytoplankton population is continuously being replaced; ii) because of their much larger surface to volume ratio, phytoplankton out-compete macroalgae in terms of nutrient uptake, the niche of macroalgae is their ability to buffer and grow on nutrients accumulated in winter when phytoplankton can't grow. We did not only compare nutrient concentrations, but also phytoplankton biomass and found no significant change. The text will be updated to clarify this. We will remove the environmental effects from the title.*

One more small detail, in chapt. 3.2 "Environmental effects" the "differences between the two reference runs" (Page 12, line 7) should rather be the difference between the reference run and the scenario with the seaweed farm application or is this a misinterpretation?

*The text is correct. Because the model does not reproduce values exactly between two 'identical' runs, only differences between the run with farm and the first reference run that exceed the difference between the two reference runs are considered significant. We will adjust the text to make this clearer.*

On the other hand the study goes beyond a simple feasibility study by providing information not only on the potential harvest that can be expected but also about the seaweed quality in relation to the use for biofuel in terms of the carbohydrate content. Especially for this "fledging industry" the relation of nutrient limitation leading to higher carbon content in context with higher product quality should be expanded into more general consideration that go beyond the individual site description. With Horizon 2020 calls on the co-use of technical structures like offshore windfarms in the North Sea for aquaculture or seaweed farms, this aspect has the potential to go beyond a pure coastal application and would raise the impact of this study towards more general consideration in this context. These additional consideration would compensate the problem that the model over- or underestimates the individual nutrients and/or chlorophyll-a concentrations at different validation or farm sites, so that it is difficult to draw more general conclusions from these aspects of the model performance.

*This is a very good suggestion, but such work should be done thoroughly using a combination of experimental and 1D modelling work to rigorously cover the relevant parameter space. The current model results would only provide a few scattered 'samples'. Work is in progress at NIOZ to start to address the effects of e.g. environmental conditions and time of harvest on carbohydrate and protein contents as well as composition in different native North Sea macroalgal species; we do not think it's advisable to use the current results for this purpose.*

When describing the nutrient validation results for Sound of Kerrera farm site (Fig. 10a and b), the question is not only on the nutrient concentration reached in comparison to the measurements. The more profound question is why the measurement show a rather different cycle compared to the simulated concentration. Is there any local source that is not reproduced in the model that brings about these differences?

*This is a good point. We have wondered about this, but do now know of one. There could be other causes such as higher local re-generation than modelled. Without additional information, we thought it best not to speculate here, and still have that opinion, so we will not make changes on this point.*

Since the mortality term is one of the key parameter for the seaweed yield a more detailed description of this process is needed. The simple hint toward "erosion" does not help the reader in which way the mortality process interacts within the simulation throughout the year. As a simple example, when at site A (Strangford Lough) the macroalgae biomass is about an order of magnitude lower than the observed one but the mortality is low throughout the farming cycles, one would interpret that the light and/or nutrient condition were not suitable in the model to efficiently reproduce the measured biomass. This implies that the mortality pressure is not an important factor when it is applied as proportional to the biomass, but this is not clear from the incomplete explanation provided.

*The model uses the same equation that relates apical frond loss exponentially to frond area as used by Broch and Slagstad. We will add a few words to this effect.*

Smaler details: For a more simple way of attributing the different sites, the characterisation A-G as used in Fig. 1 should be used as site specification next to the full locality description of the farms throughout the paper.

*We have added further references at the first use in each section.*

In this context the site specification for each farm and validation location should be highlighted in a different colour in Fig. 1 rather than simply black.

*We think that the letters and numbers provide sufficient distinction.*

In addition, since the Doggerbank and the Norwegian Trench areusedinthecharacterisationoftheNorthSeahydrodynamics,thesefeaturesshould be more pronounced in the topography map in Fig. 1, e.g. in the smaller map showing the full domain.

*We have changed the colour scale to make these features stand out more.*

Is Fig. 2 in this full detailed information overview really needed?

*Yes. Lumping all human use into one category would raise the question to provide more detail.*

The general statement on the model confirmation that "These results are not reproduced here" (page 10, line 27) should be placed on top of this subchapter.

*We cannot implement this suggestion in a form that would lead to a more legible text.*

Page 20 line 19 The first author is Jo Folden not Foden

*The text is correct.*

F Große (Referee) fabian.grosse@dal.ca

General comments The manuscript by van der Molen et al. deals with the representation of macroalgae – more specifically kelp – farms in the physical-biogeochemical model system GETM/ERSEM-BFM. Their study aims for the model-based analysis of their impact on the ecosystem as well as the productivity of such farms. A number of near-shore (four existing and one non-existing, but potentially suitable) farm sites off the British and Dutch coasts are used for this analysis. The increasing interest in macroalgae farms, which is well described in Sect. 1.1, provides a good background for this objective. Due to model limitations, e.g., with respect to spatial resolution or agreement with SPM or chlorophyll a observations, the authors clearly state that their study rather constitutes a "proof of concept" than a detailed analysis of the likely environmental impact and productivity of these farms (at end of Sect. 1.1). Despite this clearly stated limitation, I consider the objective of the study as relevant and feasible for publication. However, I have a few major and some minor points that should be addressed by the authors. My first major point of criticism is that the "environmental effects" included in the title are barely addressed in the manuscript (except for the information, that there are basically none in Sect. 3.2) and, therefore, either the title should be adapted or the manuscript should be extended with respect to that. For the relevance of the manuscript, I would propose to do the latter and will provide some suggestions in my specific comments below. Second, a partial re-ordering of the Sects. 1 and 2 (and subsections), from my perspective, would clearly improve the readability of these parts of the manuscript. Third, I would request a more detailed discussion of the model setup and the implementation of the kelp farms (and a more detailed description of the latter). For the latter, this could be especially relevant in relation to the potential environmental effects and may provide the reader with a better understanding of why the effects appear to be negligible in the present study. Provided this more detailed description and discussion of the limitations and constraints, this study may serve as a first guideline for investigating environmental effects and performance of macroalgea farms using 3D physical-biogeochemical models. Therefore, I recommend reconsidering the manuscript for publication after major revision.

Specific comments

1. Most parts of the last paragraph of Sect. 1.1 (page 3, end of line 16 to line 21) rather sound like a discussion/conclusion to me and I would suggest moving these parts correspondingly. Instead the authors could complete the short outline of the manuscript in this paragraph.

*This section describes, among others, the approach. It is important that the reader is aware of the limitations from the outset so he/she can put the results into perspective as they appear. In our opinion, this is more useful than a description of the structure of the paper, which is very standard.*

In this paragraph the authors also mention that the large-scale model allowed for the inclusion of all farms in one model (page 3, line 16). The phrasing makes it sound beneficial – which is indeed the case from a technical point of view (setting up one model instead of one for each study site). However, considering the related limitations, this should be discussed critically in the discussion section.

*We will add a line to the recommendations stating that high resolution simulations are desirable for the Norfolk farm.*

2. Organisation of Sects. 1.2 to 2.2.2: I had difficulties going through this part of the manuscript as it partly caused a back and forth reading. This mainly relates to the fact that the characteristics of the study sites (bathymetry, hydrography etc.) are provided in Sect. 1.2, followed by the description of the considered kelp species (S. latissima) in Sect. 1.3, again followed by a now more technical description of the setup of and – if existent – the sampling at the different farm sites in Sects. 2.1. I see the difficulties with the separation/combination of Sects. 1.2 and 2.1 as the former is a more general description, while the latter is more of a methodological nature. However, considering the analysis of selected study sites as part of the methodological approach, I would propose to combine the currently two subsections on the individual study/farm sites into one subsection for each farm. These subsections should then also include the representation of the individual farms in the applied model setup (e.g., position in the vertical – are all farms in the surface layer of the model?). These (general and technical) descriptions of the study sites would then be part of the methodology section (Sect. 2), preferably after the description of the implementation of the macroalgae farms to ERSEM (Sect. 2.2.2). Current Sect. 2.2.3 would then become Sect. 2.2.4. In relation to this suggestion, current Sect. 1.3 (description of S. latissima) would become Sect. 1.2. From my perspective, this is feasible as Sect. 1.1 already indicates that this study focuses on UK and Dutch sites, where S. latissima is a species of interest.

*We have adhered to the classical structure of a scientific paper in which general knowledge is summarized in the introduction, the methods are described in a methods section etc. We also think that observations should be presented before models (but have in this case put the smart-buoy and satellite data at the end so as not to interrupt the flow between the farm observations and farm implementation in the model). We recognize that there may be other ways to organize this, but adhering to the classical structure of a scientific paper is preferable and makes it easier for the reader to find the information and to know what's old and what's new.*

3. Sect. 1.2.1 (Southern North Sea): As the North Sea study sites are both located in the south-western North Sea and kelp farms will most likely be placed in the shallower southern North Sea and coastal areas (I assume, correct me if I am wrong), I would shorten the North Sea description to the parts relevant for this region, and rather extend those parts slightly. For instance, the Norwegian Trench and Skagerrak (page 3, lines 26-27) and the stratification characteristics in the central and northern North Sea (page 4, line 4) are less relevant for the context of this study, whereas the paragraph on North Sea primary production (page 4, lines 12-18) could be underpinned by some literature and more focused on the southern/coastal North Sea (those regions suitable for macroalgae farms). Differences in productivity between the Norfolk and Rhine plume sites may be stated in the subsequent paragraph (page 4, lines 20-25).

*We will remove reference to northern areas. We will add a line on differences in productivity between the Norfolk and Rhine plume sites. We will have a look at suitable references for North Sea primary production.*

4. Sect. 2.1.1. (Strangford Lough): I am no specialist in macroalgae at all, which is definitely one reason for the following two questions: The actual farm consists of 2 lines with S. latissima (that is also implemented to ERSEM) and 19 lines with other species. Though, for the model 21 lines with the former are assumed. To what extent may this affect the results? Are there strong inter-species differences, e.g., with respect to nutrient requirements? Maybe a short note on that can be made in the discussion. It is assumed "that the dry plant material consists predominantly of CH2O groups" (page 6, line 31) – is this a reasonable assumption? (Maybe underpin with literature.)

*We have only used field observations from the S. latissima long lines. We will add a sentence stating this explicitly. CH2O groups: we will add a reference to Atkinson & Smith (1983).*

5. Sect. 2.1.2 (Sound of Kerrera ...): The last sentences (page 7, end of line 10 to line 14) should be moved to the corresponding time series results or even to the discussion.

*We will remove this sentence, an equivalent statement was already included in section 3.4.*

6. Sect. 2.2.2 (Macroalgae farms in ERSEM): It is stated that the farms are implemented to the model by means of number of lines, line length etc. for which the parameter values are provided in Table 2. However, no information is provided on how the actual description/implementation in the model is done, nor is a reference provided containing such description (if existent). From my point of view this step is quite essential when presenting the study as a "proof of concept". As I could not find any other literature attempting to include macroalgae farms in a large-scale 3D model, I assume that this manuscript presents the first approach. In this case the technical description of the implementation needs to be part of the publication – not necessarily as part of the main text, but in an (electronic?) appendix or as supplementary material.

*It's simply a biomass concentration, like all other state variables in ERSEM. We will add a line to make this point more clearly.*

Regarding the model schema (Fig. 3) I also wonder whether the light climate in all parts of the water column below the farms is affected, i.e., including the below-farm parts of the grid cell with the farm itself, or only the grid cells below the farm grid cell. Depending on the vertical extent and the position of the farm, this may influence surface layer primary production by other algae (e.g., diatoms)

*Both. We will add a line to make this point more clearly.*

7. Sect. 2.2.3 (Model scenarios): For the farm scenarios, the model was run from 1 October to end of July of the subsequent year (page 9, line 33). Again a question as a non-specialist: How does this relate to the actual farming practice?

*Yes, this is a likely farming practice. We will add a statement to this effect.*

8. Sect. 3.1 (Model confirmation): Although it is probably the case, it would help to clearly state at the beginning of this section that the provided model confirmation/validation is based on the reference run.

*We will add this.*

With respect to the satellite vs. model comparison (Figs. 4and5) I wonder whether the map section could be reduced, more focusing on the regions of interest of this study (e.g.,similartoFig. 1). This would also allow for a more detailed description/discussion of the model quality in the areas of interest.

*We will change this and modify the text accordingly.*

Furthermore, it should be mentioned which months are used for the "summer", respectively, "winter" maps.

*This information was in the figure captions, but will be added to the text as well.*

With respect to the time series (Figs. 6-8), I have several comments/questions: On page 11, lines 15/16, it is stated that peak spring bloom chlorophyll a at Warp Anchorage (Fig. 6) is about 10 mg m-

3. Does this refer to the observations as the model shows much higher values (up to 30 mg m-3)? If so, this should be made clear.

*This should be: 'modelled peak spring-bloom chlorophyll concentrations were within 10 mg Chl m$^{-3}$ of the observations '. We will correct this.*

Since chlorophyll is the main quantity used in the validation, it would help to provide brief information on how the chlorophyll concentration is calculated/derived (prognostic, diagnostic using fixed/variable chlorophyll-to-carbon ratios) by the model in the model description (a reference is sufficient).

*Chlorophyll is a state variable and the model uses variable chlorophyll to carbon ratios. This information can be found in the list of references provided at the beginning of Section 2.2.2. We will add mentioning chlorophyll in that description.*

I further wonder about the large discrepancy between simulated and observed salinity and in relation to that – as also mentioned by the authors – nutrients. Does this relate to the applied river forcing? Or are there other likely causes? Although not in the focus of the study, a brief comment on that would be useful.

*This may indeed be related to the river forcing or to resolution-related issues with the representation of the river plume. We will add a remark to this effect.*

9. Sect. 3.2 (Environmental effects): As stated in my general comments this aspect is barely addressed in the manuscript. The authors mention "maps of differences in biogeochemistry and plankton dynamics" without further specifying what kind of maps. Considering, e.g., that the kelp farms affect the light availability in the deeper model layers (as indicated in the schema in Fig. 3), I wonder whether only surface maps were analysed or whether quantities in deeper layers (affected by potential changes in the light climate) were also analysed?

*This is described in section 2.2.3, with reference to an earlier paper that contains a more extensive description. The maps contained depth-averaged results, this was not mentioned and we will add this.*

At least at the farm scale, I would expect changes in the light climate in the water column below each farm. Such change may not necessarily lead to distinct changes in the biogeochemistry, due to the small farm sizes or nutrient limitation, however, it may be used as an indicator when considering an up-scaling, i.e., larger farms. Therefore, I would propose to either specify what kind of quantities were analysed without including additional results, or to show and briefly discuss the difference map of one meaningful quantity (e.g., light availability in the deeper layers during the phytoplankton spring bloom period and/or corresponding spring bloom primary production). This would strengthen the manuscript with respect to that objective of the study.

*As mentioned in section 2.2.3, we analyzed all stored model variables. We did not find significant effects for any of these, i.e. all the maps were essentially blank (give or take a few isolated spots). We would have liked to be able to present some results here, but we don't think it is useful to include and discuss a blank map… It may be possible to find short(er) periods of time that do show differences, but the objective was to look for substantial, meaningful changes, so we did not look at that level of detail.*

Furthermore, the authors refer to "differences between the two reference runs" (page 12, line 7) – based on the scenario description (Sect. 2.3) I do not understand what the second reference run is? I suppose the scenario run (incl. the farms) is referred to? This should be made clear.

*This is indeed an omission. Re-running the model with the same settings, forcings, etc. gives slightly different results. So we ran two instances of the reference run to provide a baseline difference that the scenario run needs to exceed to indicate a significant difference. We will expand the description in section 2.2.3 to reflect this.*

10. Sect. 3.3 (Strangford Lough time series): A short in-text definition of the structure-to-mass ratio (as given in the caption of Fig. 9) would be helpful.

*We will add this.*

Principally, the authors may consider defining a Sect. 3.3 "Kelp farm performance/productivity" (or similar) with the current Sects. 3.3-3.8 as subsections (then 3.3.1-3.3.6), providing a more distinct separation from the "environmental effects".

*We will adopt this suggestion.*

11. Figs. 9-13: panels k and l: What do negative uptake rates mean? Do the plots show the net uptake, i.e., uptake minus respiration? This should be clarified in the text.

*We will change the caption to 'net uptake'.*

Panels c and d: To me it is not fully clear to what the provided extinction coefficients relate – is it the one in the water column above the kelp farms or the average of the grid cell in which the farms are located? I suppose it is the former as the figure captions state "excluding contribution of macroalgae". However, it would help clarifying this in-text.

*The caption clearly states that this is at the surface.*

Similarly,Iwonderabouttheirradiance–isittheirradianceatdepthofthemacroalgae, that directly at the sea surface, or that at the centre of the grid cell? Considering, e.g., the Rhine plume farm, with a line depth of 2m and high extinction coefficients, this may result in well different values. It may help to include a brief general description of the quantities displayed in the time series at the beginning of the time series section.

*The caption clearly states that this is at the surface.*

12. Discussion/Recommendations: In relation to the authors' statement early in the introduction ("this study is a proof of concept") and the general performance/setup of the model (e.g., partly unsatisfactory reproduction of observed chlorophyll concentrations, coarse spatial model resolution), I understand that those two sections are rather general and focus on the discussion of the results on the farm performance, and provide suggestions for improved analyses of environmental impacts and farm performance. However, I would request a more detailed discussion of which limitations of the study affect the results in what way (e.g., low spatial resolution vs. small-scale environmental effects or small farm size vs. larger scale environmental effects). Part of this is already indicated during the course of the manuscript (e.g., small farm size in Sect. 3.2). Following the "proof of concept" approach of the authors, a discussion subsection dedicated to the limitations of the setup in relation to the model outcome should be provided, incl. potential effects of an improved setup, where applicable. Some of the potential limitations I raised in the previous comments. Such subsection would also provide a good basis for the recommendations section, in which the suggestions for an improved study setup can be made. From my perspective, setting up the discussion/recommendation like this would clearly strengthen the "proof of concept" aim and provide a good basis for future, more detailed studies taking into account the suggestions by the authors.

*We will insert a paragraph in the discussion on the limitations of the model.*

Technical corrections/comments 1. The authors should go through the in-text citations thoroughly and check for consistency regarding punctuation (e.g., commas before years, semi-colons between multiple citations), ordering of multiple citations (chronological or alphabetical – if I am not mistaken the journal has a preference for one of the two), in-text author names and names in the reference list (e.g., "Grosse" in-text, "Große" in the references; in "van der Molen" the V is partly uppercase, partly lowercase).

*We will address this.*

2. When providing ranges of a quantity (e.g., 15-25m) or areal extents (e.g., 5x5km) I would recommend providing the unit after both numbers.

*We will take guidance from the editor on this.*

3. Abstract: Line 2: I would rather write "and for biofuel production".

*We will change this.*

4. Introduction: Page 2, lines 26/27 and 29/30: "associated with high biodiversity is doubled. Page 4, line 13: "reduced" instead of "reducing"? Page 4, line 15: "matter" instead of "material" Page 4, line 31 to page 5, line 2: I would propose re-ordering these sentences such that there is no jump in the description from currents to depth and back to currents.

*We will change these.*

Page 5, line 4: the farm site "is" located (instead of "was")

*No, the text is correct, this farm is no longer in operation.*

Page 5, line 10: "The site range from 15-25m depth" sounds a bit odd. Maybe "The depth ranges from 15-25m at the site"?

*We will correct the text.*

Page 5, line 27: Would it be suitable to write "nitrate" instead of "nitrogen" as only nitrate uptake by kelps is considered?

*The text was correct, but we will change it into ammonium and nitrate and also modify Figure 3.*

Page 5, lines 23-30: I would change the order of the two paragraphs, as the latter is related to kelp in general while the former is region-specific.

*The first of these two paragraphs is related to the one before, so we will keep the current order.*

5. Methods Page 6, lines 25 and 30: "MPA" is used without introducing the abbreviation (line 25), while later "Marine Protected Area" is used (line 30).

*We will correct this.*

Page 6, last paragraph: I would suggest to include a sentence stating that the conversion factor used in Table 3 (24.919) results from the combination of the two factors listed in this paragraph.

*We will add this.*

Page 7, line 17: The abbreviation "ROFI" is introduced on page 3, line 11, so simply use ROFI here.

*We will change this.*

Page 7, line 29: I think, 2 digits after the comma are sufficient for the geographical information of the site.

*As this is a matter of taste, we will retain the current precision.*

Page 8, Sect. 2.2.1: Some links are shown as hyperlinks, others are not. Should be consistent.

*We will remove the hyperlink.*

Page 9, line 15: typo in "diynamics" Page 10, lines 16/17: Should the parenthesis be closed after the link, and the closing parenthesis at the end of the sentence be removed?

*We will correct these.*

Figure 2 has a rather poor quality/low resolution and the green box indicating the Norfolk kelp farm is hard to find as the map contains a lot of information. I wonder whether this map is actually necessary or if the in-text description is sufficient. If the authors prefer to keep this figure, a smaller map section may help – or addition of an arrow pointing to the farm site. In case of keeping the figure, its resolution needs to be increased.

*The map will be provided at adequate resolution. The map is necessary, and the information cannot be reduced or simplified.*

6. Results Page 12, line 21: I think it should be 0.05 kg C m-1 instead of 0.5 g C m-1. Page 12, line 24: typo in "carbohydratate"

*We will change these.*

Page 13, line 13: It is stated that Lynn of Lorne shows a slightly higher yield than Sound of Kerrera. There's a factor of 2 between most of the years that is clearly more than "slightly higher".

*We will correct this.*

Page 13, very last sentence (ending page 14): should be moved to the description of the study site and its representation in the model in the methodology section.

*This is a chicken and egg situation because we need the result of the high extinction coefficients for this. So we prefer to keep it here. It is also included in the footnotes of table 2, which is referenced in the methods section.*

Page 14, line 3: "Mortality did not increase as much as at some other sites". To my understanding this only applies to the Lyne of Lorne farm among "the other sites".

*We will remove the sentence.*

Page 14, Sect. 3.7 and Table 3: As the in-text description only refers to the wet biomass, I would display these numbers first in the table and show the harvest in parentheses or even omit the harvest and only state in the table caption that wet biomass was calculated from the harvest with reference to updated methods section (see my previous comment on the conversion factor).

*We have included both number to relate to different communities used to different units. If the editor insists we can swap the order.*

I would further omit the information on the individual Norfolk farm grid cells in the table, as the differences are quite small. Related to that the hint on the differences in the discussion could be removed (page 15, lines 22-24).

*We think that the gradient is illustrative, and may be useful if a smaller farm is implemented than simulated here.*

Figs. 9-13: Panels d: The unit of the irradiance is μmol m-2 s-1, however, in Sect. 3.8 (page 14, lines 14 and 23) μE m-2 s-1 is used. Although both are in fact the same, consistent usage of one of the two units (preferably the latter) might be helpful for the reader not too familiar with this irradiance unit.

*We thought that we had removed all instances of the μE unit, and will correct these as well.*

Panels k: Those could be named as nitrate uptake (as the parameterisation by Bloch and Slagstad only considers nitrate uptake).

*Our implementation also includes ammonium uptake, but values are likely to be small.*

Fig. 14: I would recommend re-ordering the figure panels according to the order in which the study sites are described in the text and presented in Figs. 9-13 (excl. Lyne of Lorne)

*We will change the order of the graphs.*

7. Discussion Page 15, lines 12-14: It is stated that kelp production at Rhine plume was lower than in the Sound of Kerrera and Lyne of Lorne. For the former, this is obviously not the case (see Figs. 10f and 12f).

*This is indeed an error, we will change this.*

Also, light availability and extinction are quite similar for Rhine and Sound of Kerrera sites.

*The base-line extinction at Kerrera is 0.2 $m^{-1}$, and at the Rhine site 0.5 $m^{-1}$. This is not quite similar.*

Page 15, lines 19-21: There is a factor of 2 between the production per metre between Norfolk and Sound of Kerrera, which I would not describe as "comparable".

*We will correct this.*

Page 16, line 2: "although" in combination with "however" sounds like a doubling to me. Maybe omit the latter?

*We will change this.*

Page 16, line 25: a period is missing in "eg."

*We will change this.*

Anonymous Referee #3

I am the 3rd reviewer of this manuscript and, having read the other reviews, I agree with most of the comments made by the reviewers so I am not going to repeat those, although I will highlight specific ones I find more relevant, and I will concentrate on a general evaluation. Overall, I found the article interesting and eventually deserving publication. The article is well written but I agree with Fabian Grosse's comments 1 and 2 about structuring and organisation.

*Please see our responses to Fabian's comments.*

 I have a number of general comments, below, that I think should be considered before the manuscript is acceptable for publication.
Ithasbeenhighlightedbyboththeauthorsandthereviewersthatthisislargelya"proof of concept" study and I agree with this. In that sense, I am not particularly surprised thattheenvironmentaleffectsofthemacroalgaefarmswerefoundtobenegligible,taking into account the experimental and small scale nature of all of these farms, with the exception of the "hypothetical" one off the Norfolk coast. Incidentally, the suggestion by Referee #1 of using a "comparison to the phytoplankton usually used in the model" may be an interesting way of putting their environmental effect into context.

*Please see response to comments of Referee #1.*

Another consideration is that ERSEM does not seem to do such a good job of matching observational data. This does not surprise me because I fully acknowledge the difficulties involved (e.g. due to lack of a sufficiently comprehensive forcing dataset, etc.) but it does detract somewhat from the "real-life" applicability of the present study. Therefore, I would be tempted to suggest a lesser emphasis on how the model replicates observational data (if you have those data, by all means present the comparisons but maybe just present a subset of these in the main body of the text and move the rest to supplementary material) and to dedicate more space to:

1) the technical description of the new aspects of the model (the implementation of macroalgae farms, as suggested by Fabian Grosse's comment no. 6), probably by a combination of providing a bit more detail in the main text, in addition to supplementary material;

*Strictly speaking, the model does not contain new aspects, just combinations of existing methods. The references to these should be sufficient. We will enhance the description to provide more clarity, and include a table with the parameter settings for P.*

and 2) the potential largescale development of macroalgae farming and its environmental effects. In that sense, I found the combination of a number of "real" small scale farms and a large "hypothetical" one a bit unsatisfactory; maybe the real farms should be used to illustrate how well the model fits observations (but taking into account my comments above) and a number (>1) of larger ("commercial") scale hypothetical farms could be used on a separate simulation exercise to illustrate the harvest potential (in terms of quantity and quality) of a macroalgae farming industry.

*This is exactly what is intended with the current manuscript, and this is stated in the introduction. For the hypothetical commercial farm, we have focused on a realistic possibility that is supported by regulatory bodies, rather than selecting a range of sites more or less at random, which could cause controversy. Moreover, we are not in a position to repeat the model runs within the current context and publication. This could be considered for further work.*

This relates to the comment made by Referee#1 about "more general consideration that go beyond the individual site description".

*Please see our response to the comments of Referee #1.*

This may also help address the issue of potential environmental effects, which in the present version (as identified by both reviewers) receives insufficient attention to deserve an explicit reference in the title.

*We will remove the environmental effects from the title.*

Finally, the model limitations (Fabian Grosse's comment no. 12) should also be discussed, in particular scale and resolution aspects, as well as what aspects of the model should be developed and how.

*See response to Fabian's comment.*

---

## Author Comment (AC2) · 19 Sep 2017

see pdf file with response to RC1
* * *

---

## Author Comment (AC3) · 19 Sep 2017

See pdf file with response to RC1

---

## Referee Report (RR1)

Review on "Modelling potential production of macroalgae farms in UK and Dutch coastal waters" by Johan van der Molen et al.

**General comments**

First, I would like to express that I am essentially very satisfied with the responses and changes made by the authors. They really put effort in addressing my comments on the original manuscript, which I appreciate.
Especially the inclusion of the macroalgae farm model equations and parameters I find very useful, as there are quite some differences to the work by Broch and Slagstad on which this studies bases, e.g., the inclusion of ammonium uptake. These differences are now outlined in a well understandable way.
Also, the rearrangement of the different methodology sections, from my perspective, really improved the readability and flow of the manuscript.

In principle, most of the comments I have on this revised manuscript are only of technical nature. However, I encountered a couple of inconsistencies in the new Tables 2-4 describing the farm implementation, which have to be resolved before publication. Also, I suggest to reorder some of the figures related to the sections' rearrangement.

Therefore, I recommend publication after minor revisions.

**Specific comments**

page 2, lines 17-19: The authors mention the potential of large-scale cultivation for carbon and nutrient removal/reduction. I would like to see a comment on this in the discussion – even if it is just saying that the applied setup does not allow conclusions on this. Though, for both, carbon and nutrient removal, the farm yield and C/N and C/P ratios allow for estimates which could be related, e.g., to reductions in river input in order to get an idea of their relative importance.

page 3, line 10: I would propose to remove the reference to Fig. 1 here and change the order of Figs 1-3, such that current Fig 3 comes first. This would better match with the sequence of the methods section. First: model description, second: farm sites. A reference to current Fig. 1 would be useful at the beginning of Sect. 2.2

page 5, line 33: "Table 2 to Table 4" and page 6, line 4: "see Table 5". Currently, Table 1 is referenced last (on page 8, line 21). I would suggest adding a sentence referencing Table 1 in Sect. 1.2. Maybe after the reference to "Table 1 in Kerrington et al."?

page 7, lines 22-24: "The nutrient data […] substantially lower." I would still propose to move this entire block to the results (Sect. 3.5). However, if the authors prefer to keep it here, it's fine.

page 10, lines 3-5: Would it make sense to also include relative differences for the absolute values of tidal amplitudes and currents, in order to provide a better insight into the model quality.

page 13, Sect. 3.5.3: From a "biofuel perspective", would it be worthwhile to include a few notes on the numbers of carbon extraction in-text? Perhaps, this is not too relevant for the experimental farms, but for the large Norfolk farm this would be a nice confirmation of the suitability of this site. And it would explicitly support the paragraph about the Norfolk farm on page 15 in the Discussion.

page 14, lines 4-8: the "graph" labels referring to Fig. 14 used in-text are not correct, e.g., the Rhine plume farm is graph c not a. The order in the text description should follow the order of the figure panels.

Tables 2-4:
I have a couple of comments on the equations and parameters. Some are only typesetting issues, but I also think some units (and perhaps equations) are incorrect. So, please check the tables carefully. The equation numbering I apply relates to the order of the equations in Table 4 in the revised manuscript:

1.  The type-setting of units should be consistent, e.g. "mgC" or "gC" with or without white space between "g" and "C", analogous for "gChla" and "Chl"/"Chla"; "day-1" or "d-1"; unit of "W_L" (Table 2) is probably "**m**g Chl m-2" not "**M**g …"; "-" instead of "(number)" for "n_pl" (Table 3)
2.  The units of "W_S" [in mg C m-2] and "W_C" [in gC (g sw)-1] seem to be inconsistent. Though, the denominator in the last term in the brace of Eq.1 requires this. Alternatively, some conversion factor is missing in the Eq. The same inconsistency affects Eq. 10 (e_C=…), in which the exponent needs to be dimensionless. A note on what "(g sw)-1" means would be helpful, e.g., "relative to structural mass" as in Broch and Slagstad.
3.  There is an inconsistency in the units of "A" (Eq. 2; non-dimensional) and "A0" [in m2]. This is not possible as "A/A0" is in the exponent in Eq. 3.  If "W_S" was in "mg C" this would be fine – with "A" in m2 – though, I think the issue with Eq. 1 would still remain.
4.  The unit of the exponent in Eq. 6 ("f_CL(d) = …") must be non-dimensional which is not the case as both, W_L and q_LC have "mg Chl" in their numerators
5.  In its current form, the unit of Eq. 9 (u_mn = …) should be "mmol n (mg sw)-1 d-1" as W_C is in "gC (g sw)-1"
6.  Eq. 10 (e_C + …): Even if W_C and W_S had the same unit, the exponent would not be dimensionless as y isn't dimensionless ("g C g-1"). In fact, I suspect, the units in the corresponding Eq. 15 in Broch & Slagstad are incorrect.
7.  The unit of Eq. 11 (p_L = …) is not gChl (gC)-1. Perhaps the fraction should be "W_C/W_L"?
8.  Eq. 12 (dW_N/dt = …):
    The dimensions don't match. The last term (f_CT*W_n) is not per day, probably a specific growth rate is missing. Further, W_n is in mmol m-2.

Hence, there must be a factor with unit m2 to get rid of m-2 or the entire equation has to be in "mmol m-2 d-1". μ_m is not included in any table. Do you mean μ or μ_Cm/μ_Sm?

9. Eqs. 13 & 14 (dW_C/dt and dW_L/dt): "r_C" is not defined, only "r_r" is (in Table 3). The unit of Eq. 14 must be "mg Chl d-1"

**Technical corrections**

page 3, line 2: "mixture **of** species"?

page 3, line 8: Should it be "**five** farms were simulated: **four** experimental farms"?

page 4, line 10: "species" instead of "biomass"?

page 4, line 24: please insert the type of vertical layers, before "layers" e.g., sigma or z layers?

page 5, lines 25/26: Would it be feasible to delete "as well as inclusion of and" for easier readability, without changing the meaning?

page 6, line 11: "°" is missing in geographical coordinates, while it is used in Sect. 2.1.1. It should be consistent throughout the manuscript.

page 6, lines 11-14: use either "Lough" or "lough" consistently

page 6, line 19: MLWS is only used here, so the abbreviation may be omitted

page 6, lines 20/21: already mentioned that the farm is "run by Queen's University" (on page 6, line 10)

page 7, line 26: "°" is missing in geographical coordinates

page 8, line 15: "allowing for" sounds favourable while "comparatively lower" implies the opposite. What about "resulting in lower"?

page 8, line 25: "°" is missing in geographical coordinates

page 8, line 28: use "°" instead of "degree"?

page 9, line 11: "were calculated for" instead of " were calculated of"?

page 9, line 26: Could you add a reference to current Fig. 1 for the Smartbuoy locations?

page 11pp.: Sect. 3.3 is empty. I suppose Sects. 3.4 to 3.5.4 should be subsections.

page 14, line 12: this should be "Figure **10**f"

page 14, line 14: Figure 14b does not show uptake rates. Do you mean Figure 10k,l?

page 15, line 24: "coarse" instead of "course"

page 15, lines 1 and 29: "Rhine **P**lume" vs. "Rhine "**p**lume". Should be consistent throughout the manuscript.

page 15, line 33/34: Broch and Slagstad (**2012**); same for caption of Table 4

page 16, line 17: no comma after "we do not"

Table 3:
description of "$\mu\_Cm$": unnecessary white space before "uncorrected"
descriptions of "$q\_lCS$" to "$q\_LC$": would it make sense to use "ratio" instead of "quotum"? I think it's more common but I leave it to the authors.
description of "$q\_LC$": white space missing in "perTotal"
description of "$K\_e$": typo in"whivh"
description of "$n\_pl$": "macrophyt**e**"

Table 5:
The latitudinal information for the Norfolk farm is wrong. It must be "53-something" °N. The geographical information for the other farms slightly differs from the in-text information? Is this just the difference between the real farm locations and the centres of the model grid cells in which they are located? Maybe just use one of the two consistently to avoid confusion?

Figure 1: I accept the authors' argumentation for showing the entire map, just sufficient quality/resolution of the figure should be ensured for the final publication. At the moment, it's still rather pixelated.

---

## Author Response (AR2)

Response to reviewers

Review on "Modelling potential production of macroalgae farms in UK and Dutch
5   coastal waters" by Johan van der Molen et al.

**General comments**

First, I would like to express that I am essentially very satisfied with the responses
10   and changes made by the authors. They really put effort in addressing my comments
on the original manuscript, which I appreciate.

Especially the inclusion of the macroalgae farm model equations and parameters I
find very useful, as there are quite some differences to the work by Broch and
15   Slagstad on which this studies bases, e.g., the inclusion of ammonium uptake. These
differences are now outlined in a well understandable way.

Also, the rearrangement of the different methodology sections, from my perspective,
really improved the readability and flow of the manuscript.
20   In principle, most of the comments I have on this revised manuscript are only of
technical nature. However, I encountered a couple of inconsistencies in the new
Tables 2-4 describing the farm implementation, which have to be resolved before
publication. Also, I suggest to reorder some of the figures related to the sections'
rearrangement.

Therefore, I recommend publication after minor revisions.

**Specific comments**

30   page 2, lines 17-19: The authors mention the potential of large-scale cultivation for
carbon and nutrient removal/reduction. I would like to see a comment on this in the
discussion – even if it is just saying that the applied setup does not allow
conclusions on this. Though, for both, carbon and nutrient removal, the farm yield
and C/N and C/P ratios allow for estimates which could be related, e.g., to

reductions in river input in order to get an idea of their relative importance.

*This is a good suggestion, we have added a few lines to the discussion.*

5   page 3, line 10: I would propose to remove the reference to Fig. 1 here and change
the order of Figs 1-3, such that current Fig 3 comes first. This would better match
with the sequence of the methods section. First: model description, second: farm
sites. A reference to current Fig. 1 would be useful at the beginning of Sect. 2.2

10   *The reviewer is correct that figures 2 and 3 should have been swapped following the rearrangements of the text, we have corrected this. Reference to figure 1 on page 3, however, is appropriate, and has been retained.*

page 5, line 33: "Table 2 to Table 4" and page 6, line 4: "see Table 5". Currently,
Table 1 is referenced last (on page 8, line 21). I would suggest adding a sentence
15   referencing Table 1 in Sect. 1.2. Maybe after the reference to "Table 1 in Kerrington
et al."?

*This is correct, Table 1 has been moved to the end and the tables have been re-numbered. It is not appropriate to refer to Table 1 in section 1.2.*

page 7, lines 22-24: "The nutrient data […] substantially lower." I would still propose
to move this entire block to the results (Sect. 3.5). However, if the authors prefer to
keep it here, it's fine.

25   *As the main topic of this paper is the model, it's application and results, we think it's less confusing to leave these 2 sentences where they are.*

page 10, lines 3-5: Would it make sense to also include relative differences for the
absolute values of tidal amplitudes and currents, in order to provide a better insight
30   into the model quality.

*Relative differences were not calculated as part of the publication we refer to. Doing so would require additional figures for the current manuscript that would not, in essence, provide new information, and the paper is long enough as it is. Moreover, relative differences of phase angles do not make sense. We prefer to leave this as it is.*

page 13, Sect. 3.5.3: From a "biofuel perspective", would it be worthwhile to include a few notes on the numbers of carbon extraction in-text? Perhaps, this is not too relevant for the experimental farms, but for the large Norfolk farm this would be a

5 nice confirmation of the suitability of this site. And it would explicitly support the paragraph about the Norfolk farm on page 15 in the Discussion.

*We have added a clarifying comment that what is now Table 5 contains numbers for both carbon biomass and wet biomass. It should be obvious that the carbon biomass yield is identical to carbon extraction. The additional paragraph in the*

10 *discussion (see response to earlier comment) should be sufficient to cover this point.*

page 14, lines 4-8: the "graph" labels referring to Fig. 14 used in-text are not correct, e.g., the Rhine plume farm is graph c not a. The order in the text description should follow the order of the figure panels.

*This has been corrected. In addition, Fig. 14 has been replaced with one showing the correct variable.*

Tables 2-4:
I have a couple of comments on the equations and parameters. Some are only

20 typesetting issues, but I also think some units (and perhaps equations) are incorrect. So, please check the tables carefully. The equation numbering I apply relates to the order of the equations in Table 4 in the revised manuscript:
1. The type-setting of units should be consistent, e.g. "mgC" or "gC" with or without white space between "g" and "C", analogous for "gChla" and

*Corrected*

"Chl"/"Chla"; "day-1" or "d-1"; unit of "W_L" (Table 2) is probably "mg Chl m-2" not "Mg ..."; "-" instead of "(number)" for "n_pl" (Table 3)

*Corrected*

2. The units of "W_S" [in mg C m-2] and "W_C" [in gC (g sw)-1] seem to be inconsistent. Though, the denominator in the last term in the brace of Eq.1

requires this. Alternatively, some conversion factor is missing in the Eq. The same inconsistency affects Eq. 10 ($e\_C=...$), in which the exponent needs to be dimensionless. A note on what "(g sw)-1" means would be helpful, e.g., "relative to structural mass" as in Broch and Slagstad.

*We apologize for the confusion caused by omissions and minor errors in typing the equations that have led to this set of comments. The units of* $W\_C$ *are mg C m-2, this is now corrected, and makes eqs 1 and 10 consistent.*

3. There is an inconsistency in the units of "A" (Eq. 2; non-dimensional) and
10  "A0" [in m2]. This is not possible as "A/A0" is in the exponent in Eq. 3. If "$W\_S$" was in "mg C" this would be fine – with "A" in m2 – though, I think the issue with Eq. 1 would still remain.

*The units of eq 2 were omitted, but are m2, this is now corrected.*

4. The unit of the exponent in Eq. 6 ("$f\_CL(d) = ...$") must be non-dimensional which is not the case as both, $W\_L$ and $q\_LC$ have "mg Chl" in their numerators

20  *q\_LC should be in the denominator.*

5. In its current form, the unit of Eq. 9 ($u\_mn = ...$) should be "mmol n (mg sw)-1 d-1" as $W\_C$ is in "gC (g sw)-1"

25  *Is correct after the correction in response to point 2.*

6. Eq. 10 ($e\_C + ...$): Even if $W\_C$ and $W\_S$ had the same unit, the exponent would not be dimensionless as y isn't dimensionless ("g C g-1"). In fact, I suspect, the units in the corresponding Eq. 15 in Broch & Slagstad are incorrect.

*The units of gamma are gC/gC, now corrected in Table 2.*

7. The unit of Eq. 11 ($p\_L = ...$) is not gChl (gC)-1. Perhaps the fraction should be "$W\_C/W\_L$"?

*Yes, the fraction should be W_C/W_L, now corrected.*

8. Eq. 12 (dW_N/dt = …):
The dimensions don't match. The last term (f_CT*W_n) is not per day,
probably a specific growth rate is missing.

*There should be an additional factor r_C in the last term. Also the minus sign must be a plus.*

Further, W_n is in mmol m-2.
Hence, there must be a factor with unit m2 to get rid of m-2 or the entire
equation has to be in "mmol m-2 d-1".

*This is correct, now corrected. Also for subsequent equations.*

$\mu\_m$ is not included in any table. Do
you mean $\mu$ or $\mu\_Cm/\mu\_Sm$?

*This is $\mu\_Cm$, now corrected.*

9. Eqs. 13 & 14 (dW_C/dt and dW_L/dt): "r_C" is not defined, only "r_r" is (in
Table 3). The unit of Eq. 14 must be "mg Chl d-1"

*r_r is a typo, this should be r_C. Units corrected.*

**Technical corrections**
page 3, line 2: "mixture of species"?

*Replaced by 'multiple species'.*

page 3, line 8: Should it be "five farms were simulated: four experimental farms"?

*Corrected.*

page 4, line 10: "species" instead of "biomass"?

*Corrected.*

5 page 4, line 24: please insert the type of vertical layers, before "layers" e.g., sigma or
z layers?

*This setup uses 'general vertical coordinates'. This is not the place to explain how this works. The GETM manual is available on the GETM web page.*

page 5, lines 25/26: Would it be feasible to delete "as well as inclusion of and" for
easier readability, without changing the meaning?

*Done.*

page 6, line 11: "°" is missing in geographical coordinates, while it is used in Sect.
2.1.1. It should be consistent throughout the manuscript.

*Changed.*

page 6, lines 11-14: use either "Lough" or "lough" consistently

*Done.*

25 page 6, line 19: MLWS is only used here, so the abbreviation may be omitted

*Removed.*

page 6, lines 20/21: already mentioned that the farm is "run by Queen's University"
30 (on page 6, line 10)

*Removed.*

page 7, line 26: "°" is missing in geographical coordinates

*This has been made consistent.*

page 8, line 15: "allowing for" sounds favourable while "comparatively lower" implies the opposite. What about "resulting in lower"?

*Changed.*

page 8, line 25: "°" is missing in geographical coordinates

*See earlier comments*

page 8, line 28: use "°" instead of "degree"?

*Changed.*

page 9, line 11: "were calculated for" instead of " were calculated of"?

*We have changed this to 'from'.*

page 9, line 26: Could you add a reference to current Fig. 1 for the Smartbuoy locations?

*We have added the web address.*

page 11pp.: Sect. 3.3 is empty. I suppose Sects. 3.4 to 3.5.4 should be subsections.

*This is now corrected.*

page 14, line 12: this should be "Figure 10f"

*Corrected.*

page 14, line 14: Figure 14b does not show uptake rates. Do you mean Figure 10k,l?

*We have replaced Figure 14 with the appropriate figure showing uptake rates instead of carbon structure.*

page 15, line 24: "coarse" instead of "course"

*Corrected.*

page 15, lines 1 and 29: "Rhine Plume" vs. "Rhine "plume". Should be consistent throughout the manuscript.

*Changed to lower case throughout.*

page 15, line 33/34: Broch and Slagstad (2012); same for caption of Table 4

*Corrected throughout the manuscript.*

page 16, line 17: no comma after "we do not"

*Corrected.*

Table 3:
description of "$\mu\_Cm$": unnecessary white space before "uncorrected"
descriptions of "$q\_lCS$" to "$q\_LC$": would it make sense to use "ratio" instead of
"quotum"? I think it's more common but I leave it to the authors.
description of "$q\_LC$": white space missing in "perTotal"
description of "$K\_e$": typo in"whivh"
description of "$n\_pl$": "macrophyte"

*All corrected except for the second point. We use __q__uotum related to the parameters q, as the 'r' is reserved for variables representing __r__ates of change of state variables.*

Table 5:
The latitudinal information for the Norfolk farm is wrong. It must be "53-something"
°N.

*This typo has been corrected.*

The geographical information for the other farms slightly differs from the in-text
5   information? Is this just the difference between the real farm locations and the
centres of the model grid cells in which they are located? Maybe just use one of the
two consistently to avoid confusion?

*Yes this is due to the discretisation. We have changed the caption to the table to make this clearer.*

Figure 1: I accept the authors' argumentation for showing the entire map, just
sufficient quality/resolution of the figure should be ensured for the final publication.
At the moment, it's still rather pixelated.

15   *We have removed the bathymetry from the inset and made the inset somewhat smaller.*

[revised manuscript text omitted]

---

## Author Response (AR3)

Comments to the Author:

Dear Johan,

I have read the last version of your manuscript as well as your answers to the last comments provided by one of the reviewers. Based on that I am pleased to accept your paper for publication in Biogeosciences. I have a last comment that concerns Figures 14 and 15 which do not seem to be in logarithmic scale but rather on a linear scale. Please clarify. Thank you for your efforts and for having chosen Biogeosciences for publishing your work.

Kind regards,

Marilaure

Dear Marilaure,

Thank you for your comments and for forwarding the last comments of the reviewers.

In Figure 14 and 15, the logarithm of the variable was taken and then plotted using a linear color scale. So the figures and captions are correct. This results (indeed) in a different plot from plotting the variable value using a logarithmically scaled colorbar. I agree that this could be confused, but all the information is provided properly, and I don't really see a way of making this more clearly without obviously over-doing it.

I have changed the order of Tables 5 and 6, and corrected the two mistakes in the equations.

With kind regards,

Johan

[revised manuscript text omitted]